# Solutes unmask differences in clustering versus phase separation of FET proteins

Mrityunjoy Kar [1], Laura T. Vogel [2,5], Gaurav Chauhan [3,5], Suren Felekyan [2], Hannes Ausserwöger [4], Timothy J. Welsh [4], Furqan Dar[3], Anjana R. Kamath[1], Tuomas P. J. Knowles [4], Anthony A. Hyman [1] ✉, Claus A. M. Seidel [2] ✉ & Rohit V. Pappu [3] ✉

Phase separation and percolation contribute to phase transitions of multivalent macromolecules. Contributions of percolation are evident through the viscoelasticity of condensates and through the formation of heterogeneous distributions of nano- and mesoscale pre-percolation clusters in sub-saturated solutions. Here, we show that clusters formed in sub-saturated solutions of FET (FUS-EWSR1-TAF15) proteins are affected differently by glutamate versus chloride. These differences on the nanoscale, gleaned using a suite of methods deployed across a wide range of protein concentrations, are prevalent and can be unmasked even though the driving forces for phase separation remain unchanged in glutamate versus chloride. Strikingly, differences in anion-mediated interactions that drive clustering saturate on the micron-scale. Beyond this length scale the system separates into coexisting phases. Overall, we find that sequence-encoded interactions, mediated by solution components, make synergistic and distinct contributions to the formation of pre-percolation clusters in sub-saturated solutions, and to the driving forces for phase separation.

Macromolecular condensation contributes to spatial, temporal, and functional organization of cellular matter[1–4]. As a composite process, condensation combines reversible binding, oligomerization, and coupled phase transitions such as percolation and phase separation or electrostatically-driven complex coacervation[2,5–15]. The physical chemistry of condensation has been studied extensively for FET family proteins[16]. These include the FET proteins FUS (Fused in Sarcoma), EWSR1 (Ewing Sarcoma breakpoint region 1/EWS RNA binding protein 1), and TAF15 (TATA-Box Binding Protein Associated Factor 15)[17]. FET proteins feature at least one RNA recognition motif (RRM), an intrinsically disordered arginine-rich RNA binding domain (RBD), and a prion-like low complexity domain (PLCD).

In vitro, in the presence of 150 mM KCl and at a pH of ~7.2, FET proteins purified from insect cells undergo phase separation above sequence-specific saturation concentrations ($c_{sat}$)[16]. The values of $c_{sat}$ decrease with decreasing concentrations of KCl[18]. The sequence-dependencies of measured $c_{sat}$ values have been rationalized using the framework of linear associative polymers[19], which can be used to parse the sequences of FET proteins into stickers versus spacers[16,20–24]. Stickers engage in specific interactions, and they form reversible interactions with one another. Spacers engage in weaker attractions, but they influence macromolecular solubility through effective solvation volumes also known as excluded volumes[6,23,25]. Spacer contributions to macromolecular solubility influence the balance of spacer-spacer, spacer-sticker, and spacer-solvent interactions[26–28]. Spacers also modulate the cooperativity of sticker-sticker interactions[6,23].

Phase transitions of associative macromolecules combine phase separation and percolation[5,29]. The latter is also known as

[1]Max Planck Institute of Cell Biology and Genetics, 01307 Dresden, Germany. [2]Department of Molecular Physical Chemistry, Heinrich Heine University, 40225 Düsseldorf, Germany. [3]Department of Biomedical Engineering and Center for Biomolecular Condensates, Washington University in St. Louis, St. Louis, MO 63130, USA. [4]Centre for Misfolding Diseases, Yusuf Hamied Department of Chemistry, University of Cambridge, CB2 1EW Cambridge, UK. [5]These authors contributed equally: Laura T. Vogel, Gaurav Chauhan. ✉e-mail: hyman@mpi-cbg.de; cseidel@hhu.de; pappu@wustl.edu

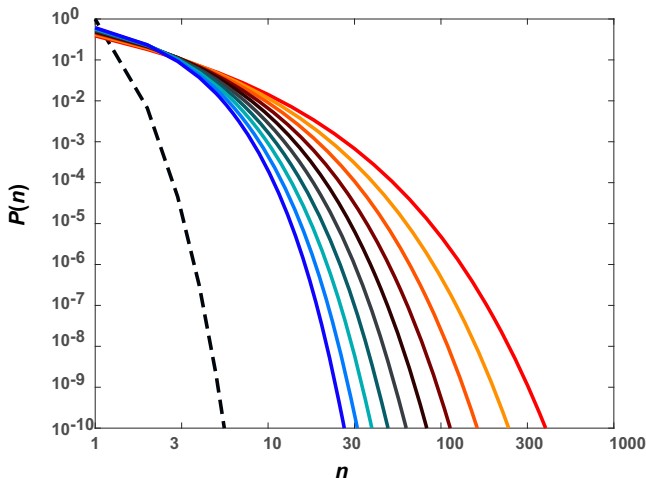

**Fig. 1 | Cluster size distributions in sub-saturated solutions for different processes.** If phase separation does not involve associative interactions and is driven by a single energy scale, *viz.*, macromolecular solubility defined by the intrinsic Flory χ parameter[40,53], then the cluster size distribution will be bounded, as shown by the black dashed line. However, if specific interactions between stickers contribute to associations, and chain segregation effects that define microphase separation[54] are absent, then the cluster size distribution evolves continuously, showing a rightward shift as $c_{sat}$ is approached – see solid lines. The ordinate quantifies $P(n)$, the probability density associated with realizing a cluster of $n$ molecules, which is the label along the abscissa. The progression from cooler to hotter colors represent increasing protein concentration.

thermoreversible gelation[19,20,30,31]. The solubility limits of macromolecules, influenced by solvent-mediated intermolecular interactions of spacers and stickers, will lead to the separation of a macromolecular solution into coexisting dense and dilute phases[32,33]. Equalization of chemical potentials and osmotic pressure[34,35] determine macromolecular concentrations in dense and dilute phases, and we designate these as $c_{den}$ and $c_{sat}$, respectively.

Multivalence, defined by the numbers of stickers of different types, will enable the networking of associative macromolecules through the formation of clusters that grow continuously with increasing numbers of molecules being incorporated into networks as concentrations increase[5,20,23,36]. These continuous transitions are defined by a percolation threshold ($c_{perc}$) above which a system-spanning network forms[5,31,36–38]. As clusters grow, their sizes will influence solubility[39]. This is because the overall solubility is governed by a combination of the sizes and physicochemical properties of clusters that form via intermolecular associations[36]. For associative macromolecules that undergo phase separation, it follows that $c_{sat} < c_{perc} < c_{den}$[5,6,20,21]. As a result, associative macromolecules featuring hierarchies of interactions undergo phase separation and percolation, and the dense phase is a physically crosslinked network that spans the condensate instead of the system[5,28,40]. This gives condensates an underlying network-like structure, which will be governed by the architectures and conformational heterogeneity of the underlying molecules. The upshot is that condensates are viscoelastic materials with sequence-specific viscoelastic moduli because the percolated network spans the dense phase[41]. Gelation or percolation sans phase separation will occur if $c_{perc} < c_{sat}$[6]. Contrary to recent assertions[42], stickers alone do not determine the driving forces for phase transitions, although $c_{perc}$ correlates positively with $c_{sat}$[16,21–23]. Instead, stickers determine the percolation threshold, spacers set the solubility limit, and synergies between stickers and spacers determine the overall phase diagram and material properties via the coupling between phase separation and percolation[5,6,25,28,30,36,41,43,44].

Distinct hierarchies of interactions that enable the classification of residues or motifs as stickers versus spacers enables the mapping of different linear heteropolymers onto linear associative polymers. These include the intrinsically disordered RGG domains of DDX4 and LAF-1[45–47], full-length FET proteins[16], their RBDs and PLCDs[26–28,43,48], the condensate driving domains of chromatin remodeling complexes[49], the stress granule protein UBQLN2[50], unfolded states of intrinsically foldable domains[24], and even RNA molecules[51].

A direct upshot of the coupling of phase separation and percolation is the presence of pre-percolation clusters in sub-saturated solutions[5,36,44]. Theory predicts that clusters grow continuously in size and abundance with increasing concentration[36,44]. Here, sizes are defined by the numbers of molecules within clusters (Fig. 1)[5,36] The distributions of cluster sizes in sub-saturated solutions will likely be heavy-tailed[5,52] (Fig. 1). This leads to a finite likelihood of forming mesoscale species, hundreds of nanometers in diameter, although their overall abundance will be low. The presence of mesoscale clusters, realized via continuous evolution of cluster sizes in sub-saturated solutions, contributes to saturating the soluble phase, thus determining $c_{sat}$. In accord with these expectations, recent studies, which deployed a diverse suite of measurements across a wide range of concentrations, showed that FET family proteins form heterogeneous distributions of pre-percolation clusters in sub-saturated solutions[52]. It was also shown that systems defined by weak clustering in sub-saturated solutions are also characterized by larger $c_{sat}$ values[52]. Above $c_{sat}$, condensate formation of FET proteins is driven by the separation of large and small species via cluster-cluster coalescence and the networking of mesoscopic clusters that form in sub-saturated solutions[52]. It is worth noting that using only one type of method such as fluorescence correlation spectroscopy (FCS) will fail to uncover the entirety of the cluster size distribution[53]. The duality of low diffusivities due to increased size and low abundance of larger clusters makes it difficult for purely FCS-based methods to detect all species present in sub-saturated solutions.

Several mutations within FET proteins were found to affect clustering and phase separation equivalently. However, separability of interactions was also demonstrated by the effects of solutes that dissolve condensates without influencing clustering in sub-saturated solutions[52]. Conversely, certain mutations can impact clustering, while having a minimal impact on phase separation, especially if $c_{sat}$ is already low. The recent study of Lan et al. reported findings for Negative Elongation Factor that support the separation of interactions that determine sub-saturated solution clusters versus condensation in live cells[54]. These observations suggest that clustering and phase separation can be governed by separable energy scales. Here, we investigated whether changing the solution anion from chloride to glutamate would have separable effects on the driving forces for phase separation versus clustering in sub-saturated solutions.

Our choice of comparing the effects of chloride versus glutamate on clustering versus phase separation was motivated by two considerations. First, glutamate tends to drive protein-protein associations[55,56]. Hence, we reasoned that glutamate should enhance clustering in sub-saturated solutions. Second, cellular milieus are complex mixtures of ions, metabolites, and osmolytes[57,58]. The high concentrations of potassium (~150 mM) are balanced by anions that include the amino acid glutamate, glutathione, and organic phosphates[58,59]. The relevant anions inside cells are glutamate and other organic phosphates, whereas the intracellular concentrations of chloride are very low in comparison (see Supplementary Table 1 for information regarding glutamate)[57,58].

There are important physicochemical differences between chloride and glutamate. For chloride, the charge of the anion is localized to the chlorine atom (Fig. 2a). Its $pK_a$ is ~ −4, making it a very strong acid or even a weak base[60]. In contrast, the $pK_a$ of the free carboxylate anion of glutamate is ~4.8, making it a weak acid (Fig. 2b).

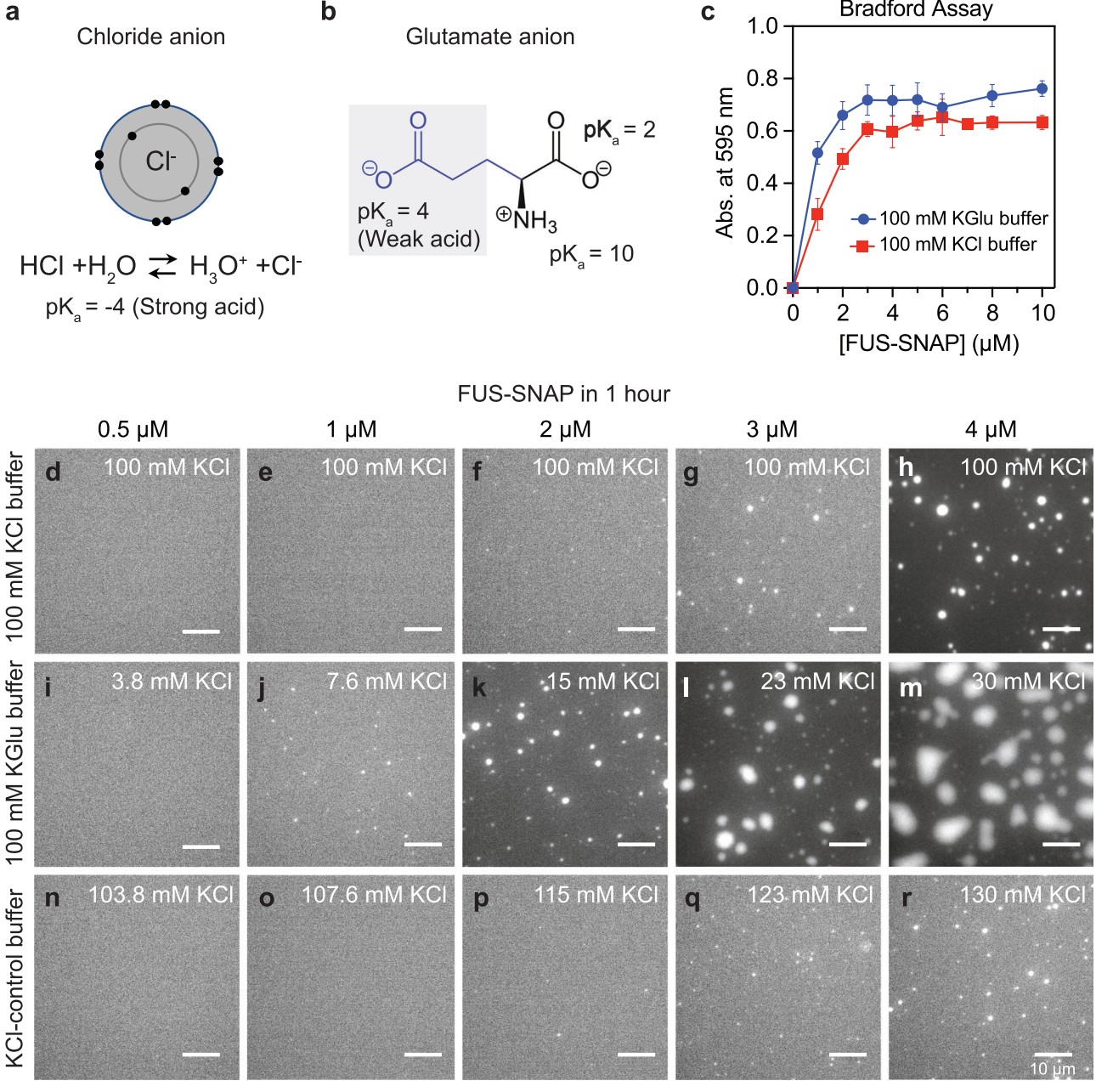

**Fig. 2 | The $c_{sat}$ of FUS-SNAP is similar in KCl versus KGlu buffers. a** Schematic of chloride anion showing its electron distribution in 2 $s$ and 2$p$ orbitals as well as its $pK_a$. **b** Schematic of glutamate. **c** Sample data for absorbance-based spin-down assays. Data are shown here for FUS-SNAP in 20 mM Tris.HCl pH 7.4, with 100 mM KCl and 20 mM Tris.Glu pH 7.4 and 100 mM KGlu at ≈ 25 °C. $n$ = 3 independent samples were used for the measurements, and data are presented as mean values ± the standard deviation (SD). **d**–**h** show bright field microscopy images collected at the 1-h time point for solutions containing different concentrations of FUS-SNAP in 20 mM Tris.HCl, pH 7.4, with a final KCl concentration of 100 mM. **i**–**m** show microscopy images collected at the 1-h time point for solutions containing different concentrations of FUS-SNAP in 20 mM Tris.Glu, pH 7.4, with 100 mM KGlu, and (**n**–**l**) show microscopy images collected at the 1-h time point for solutions containing different concentrations of FUS-SNAP in 20 mM Tris.HCl, pH 7.4, with 100 mM KCl. In both KGlu and KCl-control buffer, the residual KCl (<30 mM) from FUS-SNAP stock was added. The total concentration of KCl is marked on the (**i**–**r**). For imaging purposes, 5% of the total mixture in each sample is made up of Alex-aFlour 488 (AF488) labeled FUS-SNAP. The scale bar in each panel corresponds to 10 μm.

The charge on the carboxylate moiety is delocalized across the two $sp^2$ hybridized oxygen atoms. Accordingly, the ionic potential of Cl$^-$ is higher than the carboxylate anion, and hence chloride polarizes cations more strongly than carboxylate anions. These details have been shown to have a direct impact on the effective strengths of protein-protein and protein-nucleic acid interactions in solutions with glutamate versus chloride[55,56,61–65].

Comparative assessments of the effects of glutamate and chloride on pre-percolation clusters and phase separation of FET proteins are also motivated by recent studies on the tetramer-forming bacterial single-stranded DNA (ssDNA) binding protein (SSB)[65,66]. Kozlov et al. showed that in glutamate, as opposed to chloride, the binding of SSBs to ssDNA is highly cooperative, and this is true irrespective of the number of nucleotides that are occluded by binding to SSB[66]. This high cooperativity of ssDNA binding involves the C-terminal intrinsically disordered linker (IDL) that connects the DNA binding domain to the nine-residue C-terminal tip. The inference was that preferential exclusion of glutamate from protein surfaces is a driver of associations

of disordered regions of the SSBs[55]. These proteins also undergo phase separation in conditions that mimic bacterial milieus[67]. Kozlov et al. showed that the driving forces for phase separation, measured in terms of the temperature dependence of $c_{sat}$, was stronger in glutamate when compared to chloride[65]. For example, at 20 °C, the measured $c_{sat}$ of SSB (≈4 μM) in the presence of 40 mM potassium glutamate (KGlu) is three times lower than the $c_{sat}$ of ≈15 μM measured in the presence of 40 mM of KCl.

In this work, we compare how clustering in sub-saturated solutions is affected by glutamate versus chloride for FET proteins and specific mutants of these proteins (see sequence details in the Supplementary Data file). We use a combination of dynamic light scattering, multiparameter fluorescence detection (MFD), microfluidics-based confocal detection, and atomistic simulations to dissect the contributions of glutamate versus chloride to the driving forces for clustering in sub-saturated solutions and the driving forces for phase separation. These investigations show that interactions that drive clustering are separable from those that drive phase separation. Irrespective of the anion in solution, phase separation is driven by the achievement of a common length scale for clusters that form in sub-saturated solutions. However, the concentrations at which this length scale is realized are different in glutamate versus chloride, thus establishing how differences in the interplay between solvent-mediated and sequence-specific interactions influence clustering as opposed to phase separation.

## Results

### Enabling in vitro assessments of the effects of glutamate versus chloride

The FET family proteins cannot be purified or stored in buffers containing KGlu because the proteins precipitate under these conditions. Therefore, the proteins were purified and stored in a buffer with a high concentration of potassium chloride (KCl) (50 mM Tris-HCl pH 7.4, 500 mM KCl, 5% Glycerol, and 1 mM DTT)[52]. To prepare solutions at different protein concentrations in 100 mM KCl, the stock protein solutions were diluted and adjusted with KCl buffer. To prepare protein solutions in 100 mM KGlu, the stock solution was diluted with 100 mM KGlu buffer. Accordingly, these solutions contain an additional ~1 mM to ~30 mM KCl from the stock protein solution. To assess the effects of the residual KCl from the stock protein solutions in KGlu buffers, we performed control measurements in conditions referred to as "KCl-control buffer." This includes the corresponding amount of residual KCl from the stock protein solution and 100 mM KCl.

### Saturation concentrations of FUS change minimally in KGlu versus KCl

Using an absorbance-based spin-down assay, we quantified $c_{sat}$ values for SNAP (**S**yn**a**ptosmal-**A**ssociated-**P**rotein)-tagged FUS, designated as FUS-SNAP. The values of $c_{sat}$ were measured in 100 mM KGlu and 100 mM KCl. The spin-down assay yields an estimate of 2 μM and 3 μM for the $c_{sat}$ of FUS-SNAP in 100 mM KGlu versus 100 mM KCl, respectively (Fig. 2c).

Next, we collected microscopy images ~1 h and ~4 h after sample preparation. Results at the 1-h time point are shown in Fig. 2d–r for different concentrations of FUS-SNAP in 100 mM KCl, 100 mM KGlu, and the KCl-control buffer. In 100 mM KCl or KCl-control buffer, the formation of micron-scale condensates is detectable only at or above ~3 μM. However, in 100 mM KGlu, we observed small puncta in the concentration range of 1–2 μM. These puncta do not grow significantly over the period of 4 h (Supplementary Fig. 1). Increasing the protein concentration from 1 μM to 4 μM leads to the growth of micron-scale puncta over the 4-h window. These data suggest that glutamate has a minimal effect on the location of the phase boundary when compared to chloride. However, glutamate appears to influence the extent of clustering in sub-saturated solutions, which in turn leads to the

observed differences in kinetics for phase separation in supersaturated solutions (see data in Fig. 2d–r and Supplementary Fig. 1).

### Independent estimation of $c_{sat}$

We performed dynamic light scattering (DLS) measurements in solutions containing different amounts of FUS-SNAP to obtain an independent assessment of $c_{sat}$ (Supplementary Fig. 2). DLS probes the presence of slow modes in the temporal evolution of autocorrelation functions, and the onset of these slow modes is a feature of super-saturated solutions[52]. In 100 mM KCl, the autocorrelation functions reach a steady state below $c_{sat}$ (<3 μM) and do not change significantly over 26 min (Fig. 2d, e). Above $c_{sat}$ (Supplementary Fig. 2c), the auto-correlation functions show the presence of slow modes, which are signatures of coalescence and networking of clusters that drive phase separation[52]. In KGlu, the slow modes appear at protein concentrations of 2 μM (Supplementary Fig. 2e), whereas in KCl they appear at 3 μM[52]. These results provide independent confirmation of the estimates of $c_{sat}$ and emphasize the minor differences in driving forces for phase separation of FUS-SNAP in 100 mM KCl versus 100 mm KGlu.

### Clustering in sub-saturated solutions is enhanced in glutamate

DLS measurements show that mesoscale cluster formation of FUS-SNAP in sub-saturated solutions is significantly enhanced in KGlu buffer compared to KCl buffer (Supplementary Fig. 3). Since the scattering intensity is a convolution of a linear concentration dependence and an $R^6$ dependence on the size of the scatterers ($R$), DLS signals are dominated by a combination of the most abundant species and the largest species in solution[68]. DLS data are shown in terms of correlation coefficients. To facilitate interpretation of the correlation coefficient in terms of the sizes of mesoscale clusters, we quantified correlation coefficients for monodisperse silica nanoparticles of known concentrations and sizes (Supplementary Fig. 4). Based on these calibrations, we conclude that for a given bulk concentration of FUS-SNAP, the average sizes of clusters that form in KGlu are larger than those formed in KCl. In both buffers, the average sizes increase with protein concentration. However, the concentration-dependence of the growth of mesoscale clusters is more pronounced in KGlu versus KCl.

Next, we used nanoparticle tracking analysis (NTA)[69] to quantify the concentration dependence of forming mesoscale clusters in sub-saturated solutions (Supplementary Movie 1 and Supplementary Movie 2). NTA provides information regarding the abundance of species that contribute to the DLS signals. We collected NTA data for FUS-SNAP and untagged FUS in 100 mM KGlu and 100 mM KCl. At equivalent protein concentrations, the abundance of mesoscale clusters is ~4-fold higher in 100 mM KGlu when compared to 100 mM KCl (Fig. 3a–c and. Supplementary Fig. 5). The mesoscale clusters, which are the largest species that can form at a given level of sub-saturation, make up between ~0.2% and ~0.4% of the solution in 100 mM KCl (Fig. 3b). In 100 mM KGlu buffer, the abundance of mesoscale clusters goes up by a factor of 2-3 (Fig. 3b). We also used DLS to examine the relative abundance of mesoscale clusters in different buffers by measuring the derived count rate, which is the theoretical count rate one would obtain at 100% laser power with zero attenuation. The higher derived count rate indicates higher abundance and larger particles. Consistent with NTA data, the derived count rate for FUS-SNAP is 3-fold higher in 100 mM KGlu when compared to 100 mM KCl (Fig. 3d). Untagged FUS shows similar behaviors (Supplementary Fig. 5g).

To assess the transferability of our findings to other members of the FET family, we performed DLS measurements and quantified derived count rates for FUS-EGFP (Supplementary Fig. 5h), TAF15-SNAP (Fig. 3e) and EWSR1-SNAP (Fig. 3f). In all systems, we observe a similar trend, where the derived count rate is higher in 100 mM KGlu when compared to 100 mM KCl at each of the protein concentrations studied. The protein concentration of TAF15-SNAP in its stock solution

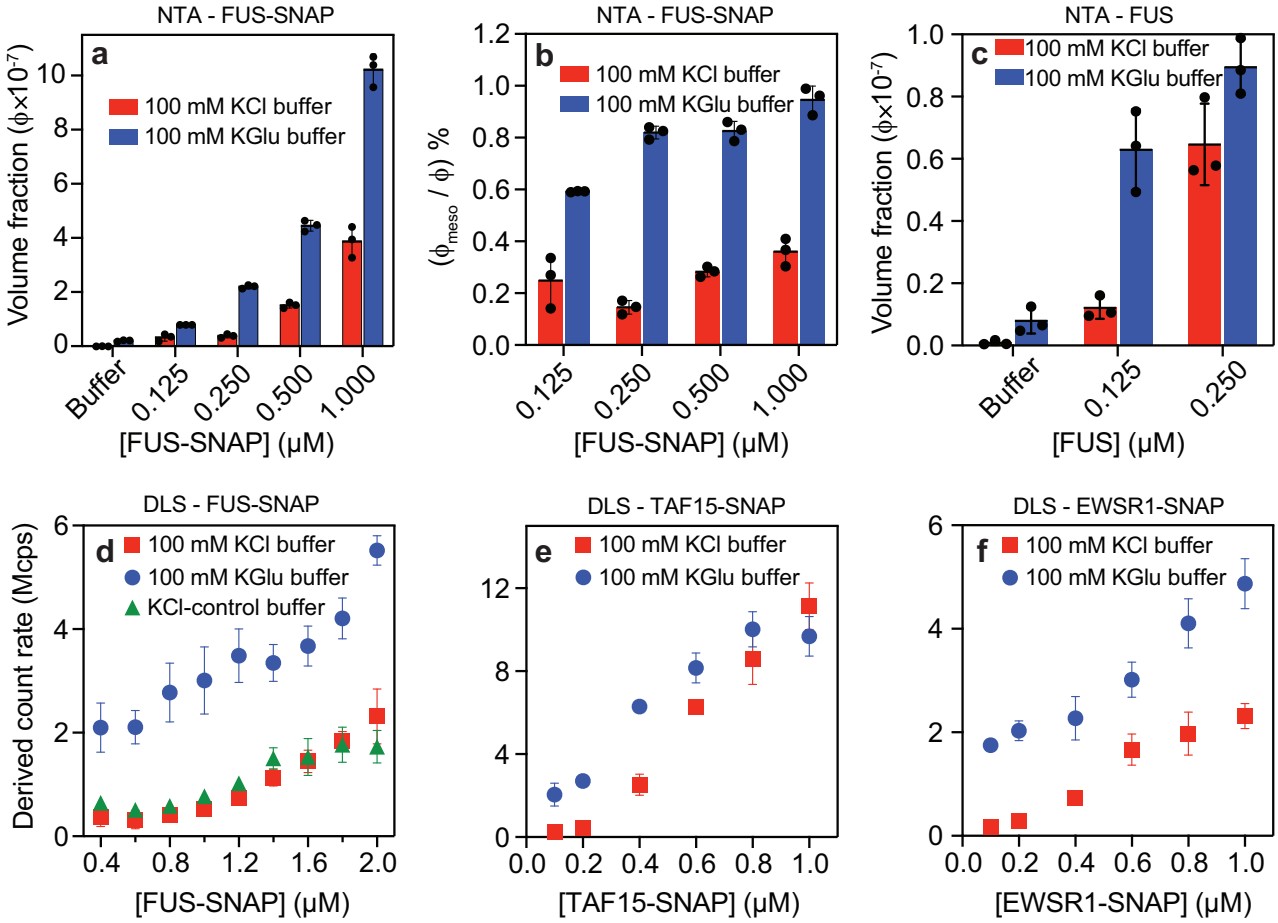

**Fig. 3 | Abundance of mesoscale clusters in sub-saturated solutions of FET proteins increases in KGlu compared to KCl buffers. a** NTA data show the volume fraction of mesoscale clusters for FUS-SNAP at different sub-saturation concentrations in 100 mM KCl and KGlu buffers. **b** Relative abundance of mesoscale clusters for FUS-SNAP at different sub-saturation concentrations in 100 mM KCl and KGlu buffers. **c** NTA data help with quantifying the volume fraction of mesoscale clusters for FUS at different sub-saturation concentrations in 100 mM KCl and KGlu buffers. DLS data show the derived count rates of FUS-SNAP (**d**), TAF15-SNAP (**e**), and EWSR1-SNAP (**f**) in KGlu versus KCl buffers. $n = 3$ independent samples were used for the measurements, and data are presented as mean values $\pm$ SD.

is low (~12 μM). Accordingly, with increasing protein concentrations, the method of sample preparation leads to higher residual KCl in the solution. Therefore, at 100 mM KGlu buffer, the residual KCl suppresses some of the clustering, resulting in a lower differences in the derived count rate than in 100 mM KCl. Similar effects were also observed for untagged FUS (Supplementary Fig. 5g).

### Size distributions of low abundance mesoscale clusters from analysis of DLS data
The mesoscale clusters represent 0.1–1% of all species that are present in sub-saturated solutions. The abundance of mesoscale clusters is higher in glutamate than in chloride, especially well below $c_{sat}$ (Fig. 3a–c). We used the number density of scatterers extracted from the DLS data and quantified the distribution of hydrodynamic diameters ($d_H$) of mesoscale clusters. We used this analysis to ask and answer a set of questions. On the manifold of mesoscale species that are of low overall abundance, viz., the largest species that form in sub-saturated solutions, what is the frequency of observing specific $d_H$ values? Is there evidence for the continuous evolution of the tail in the cluster size distribution as depicted in Fig. 1, and is this evolution different in KCl versus KGlu? To answer these questions, we leveraged the fact that information regarding the time correlation functions combined with information regarding raw intensities can be used to extract distributions of scattering intensities, which scale as the sixth power of the hydrodynamic diameter $d_H$. Using the Stokes-Einstein

formula, these intensity distributions can be used to estimate the number densities of $d_H$ values. Following the approach of Cohan et al. [70], the intensity distributions were converted to distributions of $d_H$ values using practical implementations of Mie scattering theory[68].

We extracted distributions of $d_H$ values for FUS-SNAP at different protein concentrations. All measurements were performed in sub-saturated solutions. We compared the distributions in 100 mM KCl versus 100 mM KGlu (Fig. 4a–d) at different protein concentrations. For three of the four protein concentrations (0.125 μM, 0.25 μM, and 0.5 μM) the size distributions in KGlu are shifted to higher values when compared to KCl. The distributions in KGlu and KCl show the heavy tail nature that we anticipate from theory (Fig. 1). This point becomes clear when one accounts for the abundance of the mesoscale clusters, which as measured using NTA shows that mesoscale clusters make up 0.1–1% of the solution. Interestingly, the continuous growth of mesoscale species with increased protein concentrations plateaus in KGlu as $c_{sat}$ is approached. As a result, in a 1 μM protein solution, the distributions of $d_H$ values in KCl catch up with the distributions in KGlu. This observation helps explain the similarities of $c_{sat}$ values that we estimated in KCl and KGlu. It is also in line with the theories[39,71], where the entropy of mixing becomes considerably less favorable as molecular or cluster sizes increase—a phenomenon referred to as an entropic sink by Bracha et al.[72]. The implication is that there exists a length scale in terms of cluster size, which when crossed, enables the separation of the solutions of FUS molecules into coexisting phases.

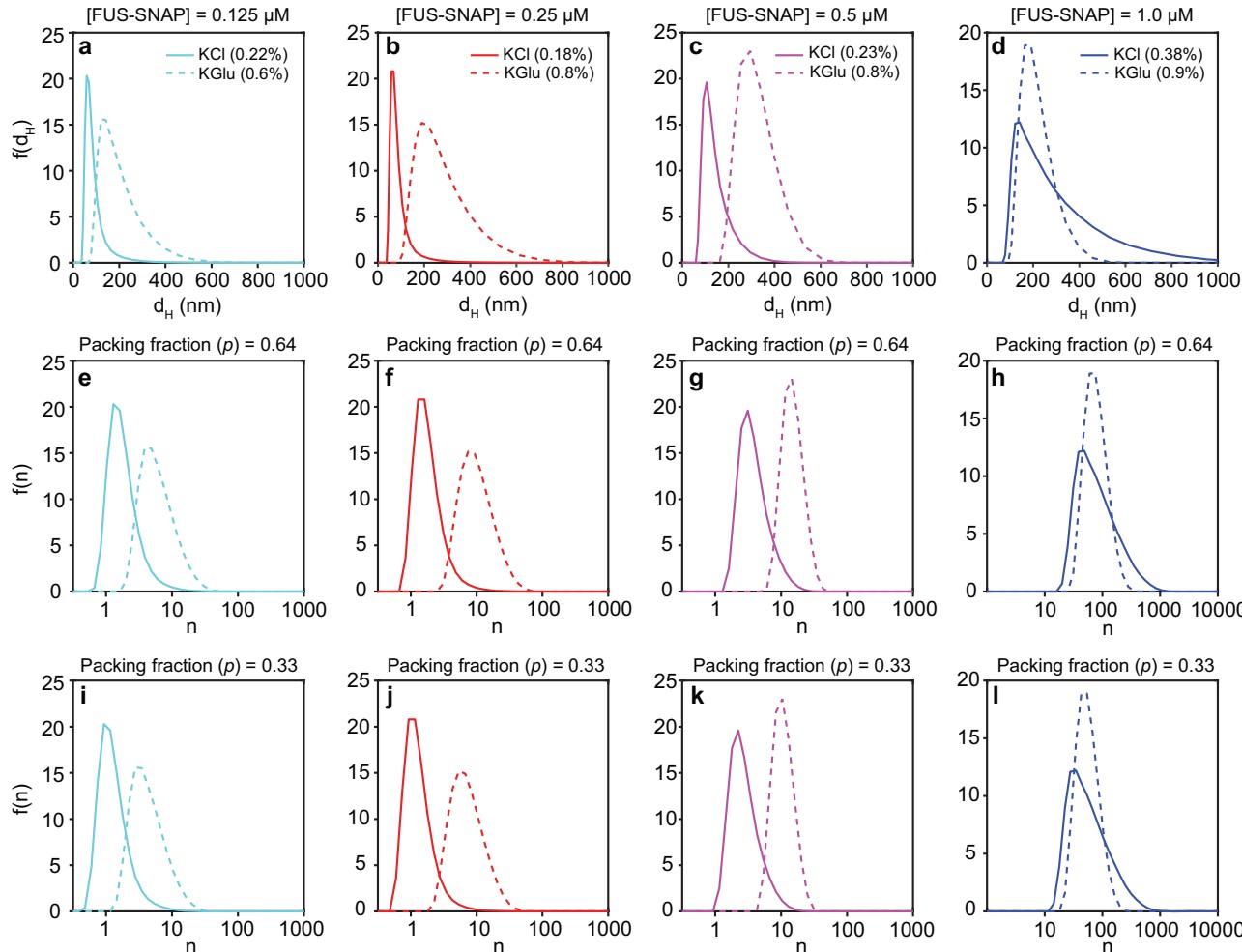

**Fig. 4 | Size distributions of low abundance mesoscale clusters.** We extracted the distributions of $d_H$ values for FUS-SNAP at different protein concentrations, all of which were in the sub-saturated regime. The top row shows the distribution of $d_H$ values extracted in KGlu (dotted curves) and KCl (solid curves) for solutions with protein concentrations of 0.125 μM (**a**), 0.25 μM (**b**), 0.5 μM (**c**), and 1 μM (**d**). These distributions are shown as raw histograms, and hence the ordinate shows frequencies i.e., the number of occurrences of a $d_H$ value between $d_H$ and $d_H + \Delta$, where $\Delta = 0.1$ nm. In each panel, the abundance of species being analyzed is shown in the legend, and these values were extracted from NTA data shown in Fig. 3b.

Rows 2 and 3 show the distributions of the number of molecules n within a cluster of size $d_H$. These distributions were computed by assuming that the molecules within clusters are spheres. The packing fraction can be set to be $p = 0.64$, for random close packing of spheres, (**e–g**) or $p = 0.33$, (**h–i**), assuming a packing density concordant with reports of the volume fractions of protein versus solvent in single protein condensates[26–28]. In each panel, the solid curve corresponds to KCl, and the dotted curve corresponds to KGlu. The concentration of [FUS-SNAP] for each column of plots is shown at the top.

Next, we converted the distribution of $d_H$ values to estimate the cluster size distributions. We report these distributions as frequency histograms since the mesoscale clusters are the least abundant species in solution and normalizing the histograms to compute $P(n)$ as in Fig. 1 would require knowledge of the full species distribution. To extract the frequency histogram of cluster sizes, we used the fact that the $d_H$ of monomeric FUS-SNAP is 4.6 nm (see below for a direct measurement). Assuming a spherical approximation for the monomers, the number of molecules $n$ within a cluster of hydrodynamic diameter $d_H$ can be estimated using $n = p(v_c/v_m)$. Here, $v_c$ and $v_m$ are the volumes of the cluster and monomer, respectively and $p$ is a dimensionless packing fraction. The upper limit on $p$ is 0.74 for crystalline packing of monomers within a cluster. If we assume random close packing of spheres, then $p = 0.64$. And if we assume that molecules are packed within clusters as they would be in dense phases, where the volume fraction of solvent is between 0.6 and 0.7[26–28,43,73], then we can set $p = 0.33$.

We analyzed the distributions of cluster sizes for mesoscale clusters formed by FUS-SNAP by assuming two different values for $p$ namely, 0.64 (Fig. 4d–g) and 0.33 (Fig. 4h–k). These choices ignore the possibility that the packing fraction might depend on the cluster size.

Both choices for $p$ paint a similar picture. The low abundance mesoscale clusters, which should be in the tails of the cluster size distribution, show a rightward shift toward larger numbers with increasing protein concentration. The cluster sizes in KGlu are larger than in KCl for three of the four concentrations. As $c_{sat}$ is approached, the cluster sizes stop growing in KGlu, and the cluster size distribution in KCl becomes akin to what is observed in KGlu.

When comparing these cluster size distributions to what we anticipate from theory (Fig. 1), it is important to remember that we are analyzing cluster sizes on the manifold of mesoscale species whose abundance is low, being in the range of 0.1–1%. The implication is that the DLS data are probing the most abundant species, and the tails of the cluster size distributions but nothing in between. To go beyond the tails and obtain information regarding the totality of size distributions, we used multiparameter fluorescence measurements.

### Detection of the full range of species that form as a function of protein concentration

To extract the size distributions and the concentration-dependent evolution of species distributions that are smaller than mesoscale

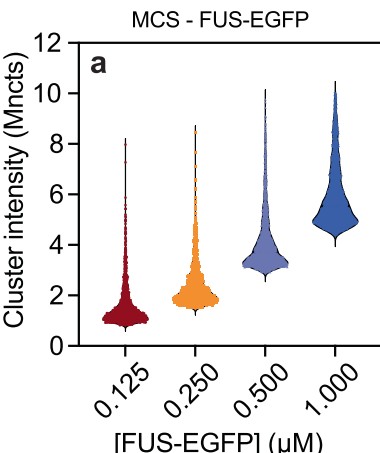
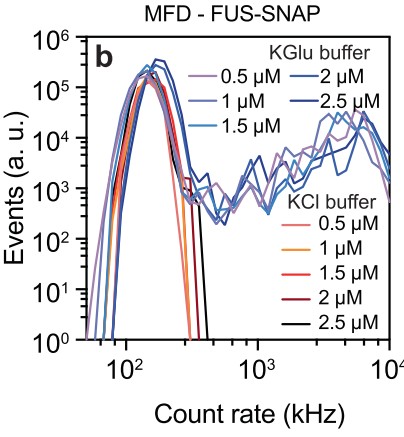
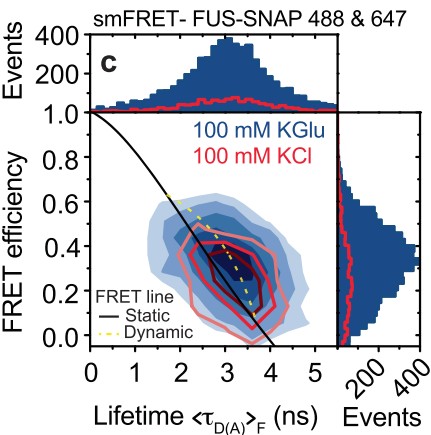

**Fig. 5 | Multiparameter fluorescence measurements show continuous concentration-dependent evolution of clusters that are different in KGlu versus KCl. a** Single-molecule analysis by microfluidic confocal spectroscopy (MCS) shows that FUS-EGFP cluster formation increases with increasing concentrations from 125 nM to 1000 nM in KGlu buffer. **b** Intensity distributions from the multiparameter fluorescence detection (MFD) experiment in the presence of 15 nM Nile red, measured at various concentrations of FUS-SNAP, show that the abundance and sizes of clusters increase in 100 mM KGlu. This is evidenced by the tailing towards higher count rates up to $10^4$ kHz per burst compared to 100 mM KCl buffer. **c** Single-molecule FRET measurements with 200 pM of FUS-SNAP-AF488 as donor and FUS-SNAP-AF647 as acceptor in KGlu buffer (blue) compared to KCl buffer (red) show that the number of FRET events are significantly higher for KGlu (1D-histograms). The FRET populations are broadened beyond shot noise due to intermolecular dynamics (dynamic FRET line, dashed light green), resulting in a deviation from the static FRET line (solid black).

clusters, we used microfluidic confocal spectroscopy (MCS) to investigate the evolution of clusters of all sizes that form in sub-saturated solutions in the presence of glutamate. MCS is a brightness-based method that combines microfluidic mixing and flow with confocal detection[74]. As with previous studies, which were performed in the presence of 100 mM KCl[52], we observed a monotonic increase in the brightness per molecule for FUS-EGFP in 100 mM KGlu where EGFP refers to enhanced green fluorescent protein. Clearly, clusters form in sub-saturated solutions, and their average sizes increase with increasing concentrations (Fig. 5a).

Building on the MCS measurements, we used MFD as a complementary method that affords access to distributions of clusters that form in the presence of glutamate versus chloride. These measurements are sensitive to small and intermediate sized species, and they provide a complement to the measurements of mesoscale clusters extracted from DLS and NTA. We used increasing concentrations of FUS-SNAP stained with a fixed concentration (15 nM) of Nile Red in the presence of either 100 mM KGlu or 100 mM KCl. The fluorescence intensity distributions, which were extracted using analysis of fluorescence bursts, showed clear and pronounced differences between glutamate versus chloride (Fig. 5b). Additionally, in glutamate, we observed a continuous tailing towards count rates of up to $10^4$ kHz. In contrast, we did not observe significant populations beyond 400 kHz for FUS-SNAP in KCl. These results suggest that clustering in sub-saturated solutions is enhanced in KGlu versus KCl.

Fluorescence intensity distribution analysis (FIDA) provides insights regarding the growth in brightness, and this is a useful proxy for the sizes of protein clusters. Inferences regarding the sizes of protein clusters can be drawn by analysis of donors and acceptors separately (Supplementary Fig. 5). The elongated shape of the contours in two-dimensional histograms (Fig. 5c) suggests the formation of dynamic clusters of FUS molecules that rearrange on the millisecond timescale. This is supported by clear deviations from the static FRET (Förster resonance energy transfer) line. Assuming a simple structural model consisting of two states, the datasets from both KCl and KGlu buffers can be fitted with the same dynamic FRET line[75]. The FRET efficiencies show similar averages in both buffers indicating that the differences in anions do not induce detectable structural changes. Measurements of FCS show that the autocorrelation of donor-labeled

FUS-SNAP is consistent with increased translational diffusion time $t_{d2}$ and strong fluctuation in the weighted residuals at correlation times longer than $t_{d2}$ in the KGlu buffer (Supplementary Table 3). This points to the presence of higher-order complexes even at 400 pM concentrations of FUS-SNAP. These single-molecule data show clear evidence for increased heterogeneity and larger cluster sizes for FUS-SNAP in glutamate versus chloride.

## FCS and NanoDSF show enhanced stabilization of FUS clusters in glutamate

IUPRED analysis[76] predicts that isolated FUS mainly consists of disordered regions (Fig. 6a). However, given extant sequence-ensemble characterizations of disordered proteins[77], it stands to reason that there will be conformational fluctuations that are differently impacted by glutamate versus chloride. To investigate the influence of both buffers on the stabilization of FUS in monomers and in clusters, we analyzed conformations and associations of FUS-SNAP-AF488 in a complementary approach by FCS at the single-molecule level and by nanoscale differential scanning fluorimetry (nanoDSF) at concentrations close to saturation where the signals will be dominated by larger, non-monomeric species (Supplementary Fig. 7).

We studied single-molecule events in equilibrated solutions with FUS-SNAP-AF488 in KGlu and KCl, respectively (Fig. 6b). By comparing the signal traces, we see that the bursts in KGlu are brighter and more frequent. We computed the autocorrelation functions of FUS-SNAP-AF488 for two intensity-based selections namely, monomers and clusters. For the preferential selection of monomers, all bright bursts above its intensity threshold were excluded. In Fig. 6c, we show correlation curves for FUS monomers together with the free dye measurement of rhodamine 110 as a reference. The data for FUS were fit using a model (see Methods) with two components for translational diffusion namely, one global time for dye impurities and one salt-dependent time for monomeric FUS. In the panels on the right, we show two blow-ups of the correlation curves centered on the respective diffusion times of FUS monomers, $t_{d,monomer}$, (dark yellow) and clusters, $t_{d,cluster}$ (pink). For FUS monomers, the correlation curves in KCl and KGlu overlap, and the fitted diffusion times are identical within error. Using the Stokes-Einstein equation, we converted these times to an average hydrodynamic radius of 2.3 nm.

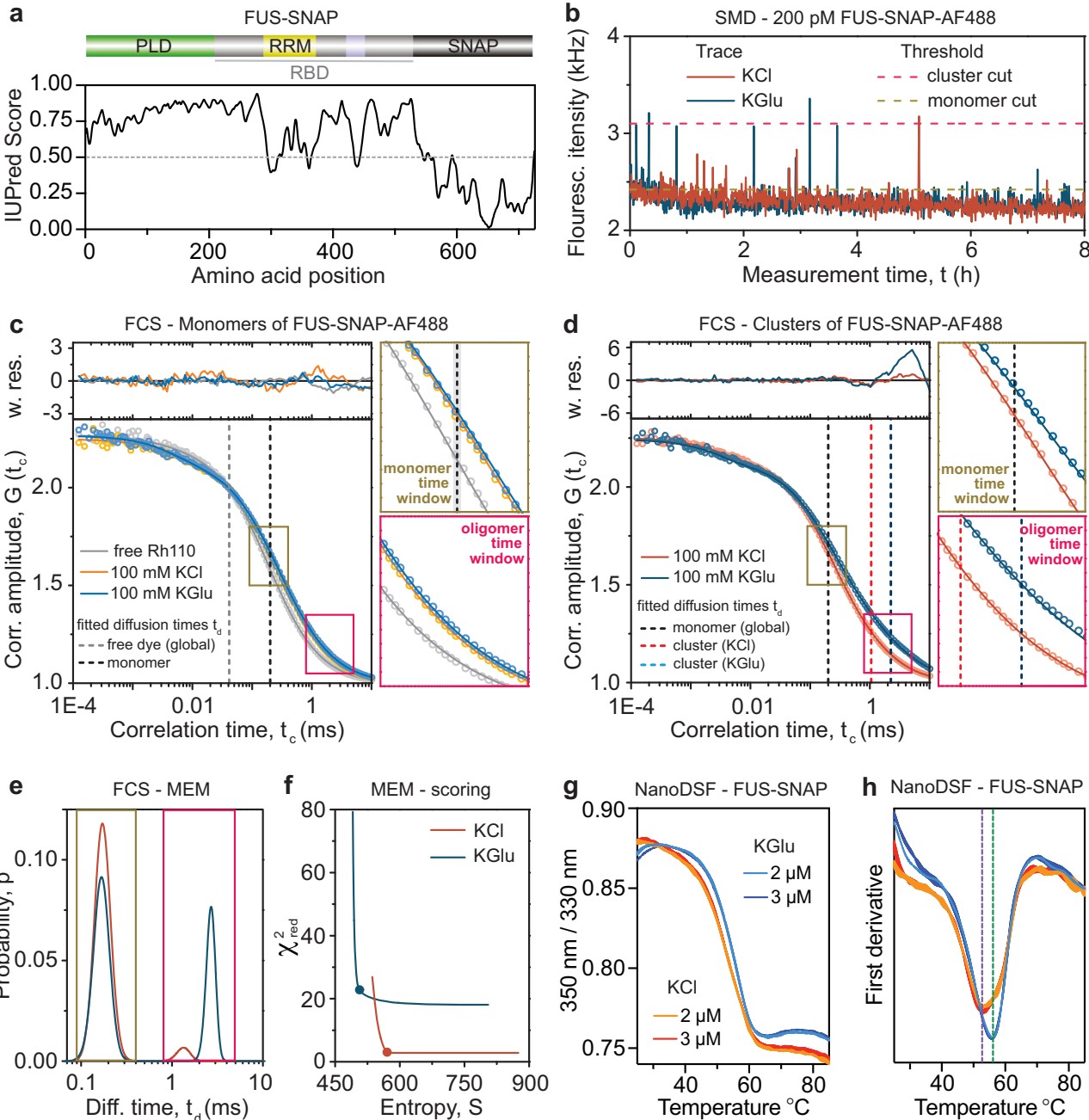

**Fig. 6 | Compared to KCl, KGlu stabilizes FUS-SNAP clusters even at ultra-low concentrations. a** IUPRED predictions of disorder of FUS-SNAP. **b** Single-molecule detection (SMD) fluorescence intensity traces of 200 pM FUS-SNAP-AF488 in both buffers with an indicated threshold for cluster cut (yellow-green) and monomer reference cut (pink). These traces display the pronounced clustering behavior in KGlu compared to KCl. **c**, **d** Autocorrelation curves for FUS-SNAP-AF488 with blow-ups for the monomer time window (yellow-green) and oligomer time window (pink) show the resulting translational diffusion times (dashed lines), including one global and buffer-dependent time for a 3D-Gaussian diffusion model and the weighted residuals for the cluster cut and the monomer cut as reference. The fit to Eq. 6a (see Methods) in (**c**) yields identical diffusion times within error in KCl ($t_{d,mo}^{(KCl)}$ = 0.189 ± 0.017 ms, orange) and KGlu ($t_{d,mo}^{(KCl)}$ = 0.201 ± 0.017 ms, and blue (for all fit results, see Supplementary Table 3). Additionally, the free dye measurement of Rhodamine (Rh110) is given as reference (gray) in (**c**). The correlation curve for

the cluster cut displays in KGlu (dark blue) a clear shift to longer diffusion times in the oligomer time window when compared to KCl (red). The monomer component (dashed black) is adequately fitted with one diffusion time for both buffers (see Supplementary Table 4). **e** The maximum entropy method (MEM) gives diffusion time distributions (101 components) as a function of probability with one peak between 0.09 and 0.4 ms (monomer time window, green) and a second peak between 0.8 and 4 ms (oligomer time window, pink). **f** Corresponding L-curves, according to ref. 79, are presented as quality validation for the obtained diffusion time distributions where the corner point is indicated by a circle. **g**–**h** NanoDSF data show the ratio of 350 nm/330 nm plotted against temperature for FUS-SNAP in KCl and KGlu buffers. **h** The first derivative of data shown in (**g**) against temperature shows the apparent unfolding temperature of FUS-SNAP at 57 °C and 53 °C in KGlu and KCl buffers, respectively. $n$ = 3 independent samples were used for the measurements, and data are presented as mean values ± SD.

Therefore, the distinct buffers do not change the overall size of monomeric FUS.

In contrast to the monomer sub-population, the long diffusion time $t_{d,cluster}$ (Fig. 6d) of FUS clusters in KGlu differs from the value in KCl by ~1 ms (Supplementary Table 4). Furthermore, large deviations in the weighted residuals indicate the insufficiency of a two-component fit for FUS clusters in glutamate. Thus, we applied the Maximum Entropy Method (MEM) as a model free approach[78,79] to quantify the diffusion time distributions for clusters (Fig. 6e). Two peaks were obtained for both buffers. To verify the goodness of the fit and demonstrate appropriate weighting, we display the dependence of the reduced $\chi^2_{red}$ on the entropy (L-curve, Fig. 6f), where the chosen values of the corner point are marked with a dot (see Methods and Supplementary Fig. 6). Due to the larger fraction of clusters, a higher minimum $\chi^2_{red}$ is yielded for KGlu. The first peak at $t_{d,monomer}$ ~ 0.2 ms resembles the monomer species and they overlap for KCl and KGlu. The second peak, which is in the millisecond time range, corresponds to clusters. In KGlu, the peak has significantly longer times and higher amplitudes than in KCl. From this we conclude that FUS clusters are more abundant and larger in size in KGlu, even though there are no detectable conformational differences at the level of FUS monomers. Instead, glutamate enhances macromolecular associations when compared to chloride, and this is in line with the previous reports[61,66].

Glutamate is known to enhance protein stability[56]. Although FUS is intrinsically disordered, its overall dimensions and the heterogeneity of intramolecular interactions, quantified via accessibility of different functional groups, will be temperature dependent. Accordingly, we investigated how buffers influence the temperature dependence of tryptophan fluorescence of FUS-SNAP. For this, we performed nanoDSF measurements, which helps us analyze the consequences of thermal fluctuations in low-volume capillaries. Increasing the temperature will drive increased exposure of tryptophan residues. NanoDSF monitors the concurrent changes in tryptophan fluorescence at 330 and 350 nm[80]. To increase the measurement sensitivity, we used FUS-SNAP instead of FUS. This helps minimize the amount of residual KCl caused by the storage buffer, and it enables improved signal strength.

Figure 6g shows changes of the 350 nm/330 nm ratio as a function of increasing temperature in two different concentrations and buffers. The first derivative plot (Fig. 6h) shows that the apparent unfolding temperatures of FUS-SNAP in the KGlu and KCl buffer are 57 °C and 53 °C, respectively. We also tested the SNAP-tag alone as a control. The apparent unfolding temperature of SNAP in the KGlu and KCl buffer is 65 °C and 69 °C, respectively (Supplementary Fig. 6). Surprisingly, in the KGlu buffer, SNAP has lower apparent unfolding temperature than in the KCl buffer. Therefore, the enhanced thermal stability of FUS-SNAP in glutamate can be attributed to FUS and not to SNAP. Taken together, the FCS and nanoDSF measurements demonstrate that glutamate enhances intramolecular and intermolecular interactions among FUS molecules when compared to KCl.

### Differential anion effects probed using mutations reveal the origins of separation of energy scales

We investigated the effects of mutations within the FUS PLCD on clustering in sub-saturated solutions in the presence of 100 mM KGlu versus 100 mM KCl buffer. Tyr-to-Ser mutations increase $c_{sat}$ by up to two orders of magnitude compared to wild-type FUS[16]. Mesoscale clusters are undetectable using DLS in either 100 mM KCl or 100 mM KGlu. However, some level of clustering is evident when 18 or 10 Tyr residues in the PLCD are substituted to Ser in the presence of 100 mM KGlu (Fig. 7a). These results implicate Tyr residues in PLCDs as being important for clustering and phase separation. Substitution of 24 Arg residues within the RBD to Gly also increases $c_{sat}$ by up to two orders of magnitude compared to wild-type FUS[16]. These substitutions abrogate clustering in sub-saturated solutions in both chloride and glutamate

(Fig. 7b). Taken together with the results from substitutions of 27 Tyr to Ser in the PLCD, the substitutions of Arg to Gly in the RBD emphasize the importance of networks of Tyr-Arg interactions as drivers of clustering in sub-saturated solutions and of phase separation as well.

Substituting 24 Arg residues in the RBD to Lys increases $c_{sat}$ by an order of magnitude compared to wild-type FUS[16]. Strikingly, while these substitutions weaken clustering in the presence of 100 mM KCl (Fig. 7c), the extent of clustering we observe in the presence of 100 mM KGlu is akin to that of wild-type FUS (compare Fig. 7c to Supplementary Fig. 5g). As shown in recent single-molecule studies, there is a uniform weakening of cation-π interactions in chloride salts[81]. In contrast, in glutamate, the differences between wild-type FUS and the 24R-K variant seem to be length-scale dependent. Specifically, while clustering is preserved upon substituting Arg to Lys, phase separation, which should be governed mainly by solubility, is weakened by Arg to Ly substitutions. This can be rationalized if the strengths of cation-π interactions are minimally affected by glutamate compared to chloride. However, the increase in $c_{sat}$ points to differences in solubility driven by substitutions of Arg to Lys. Indeed, it is noteworthy that the free energies of solvation of Arg and Lys are fundamentally different from one another[82], and Arg has more of a hydrophobic character than Lys[82]. As a result, the driving forces for phase separation, which are governed by solubility limits, are affected by substitutions of Arg to Lys, whereas clustering in sub-saturated solutions is not affected, especially in glutamate.

The importance of multivalent interactions on clustering in sub-saturated solutions is also underscored by the effects of substitutions of Tyr or Phe residues within the RBD to Ser or Gly, respectively. In both chloride and glutamate, these substitutions weaken the extent of clustering in sub-saturated solutions (Fig. 7d). Finally, substitutions of 10 Asp and 4 Glu residues within the RBD significantly enhance clustering compared to wild-type FUS, and the effects of these substitutions are similar in glutamate and chloride (Fig. 7e). Note that while clustering is weakened and its detection requires the use of 10 μM protein for the 6F-G and 6Y-S variants (Fig. 7d), clustering is readily apparent at 250 nM of the 10D/4E-G variant (Fig. 7e).

### Clusters form reversibly in sub-saturated solutions

The average sizes of clusters in sub-saturated solutions from DLS experiments show a decrease upon dilution and an increase with increased concentration (Fig. 8a, b). This is true in glutamate and chloride buffers[52]. Similarly, in MFD experiments (Fig. 8c), when 3 μM of Nile Red stained FUS-SNAP was diluted to 1 μM, the number of bursts decreased. Increasing the protein concentration to 2.75 μM also increased the bursting, and upon dilution to 1.65 μM, the number of bursts decreased. For each protein concentration, the increases in fluorescence amplitude and broadening of the fluorescence distributions become less significant at later dilution steps, indicating the reversibility of FUS clusters. In smFRET studies, the introduction of 40 nM of unlabeled FUS-SNAP causes a shift in the FRET signal to very low FRET between FUS-SNAP-AF488 and FUS-SNAP-AF647 (Supplementary Fig. 8a). This shows that cluster formation is reversible and that molecules within clusters are labile in glutamate and chloride[52].

### Clusters create distinctive local environments

Next, we used environmentally-sensitive dyes, specifically Nile red and bis-ANS, to probe the local environments of clusters that form in different buffers. The quantum yields of both dyes increase in nonpolar environments[40,83,84]. We measured the fluorescence lifetime distributions of Nile red using MFD in various concentrations of FUS-SNAP in 100 mM KGlu and 100 mM KCl. Nile Red exhibits a spectrum of lifetimes ranging from 0.6 ns to 4.66 ns. It is known the lifetimes of Nile Red increase with increased hydrophobicity[85]. In 100 mM KCl, at 0.5 μM FUS-SNAP, the peak in the fluorescence lifetime distribution occurs at 2 ns. With increasing concentration of FUS-SNAP, the lifetime

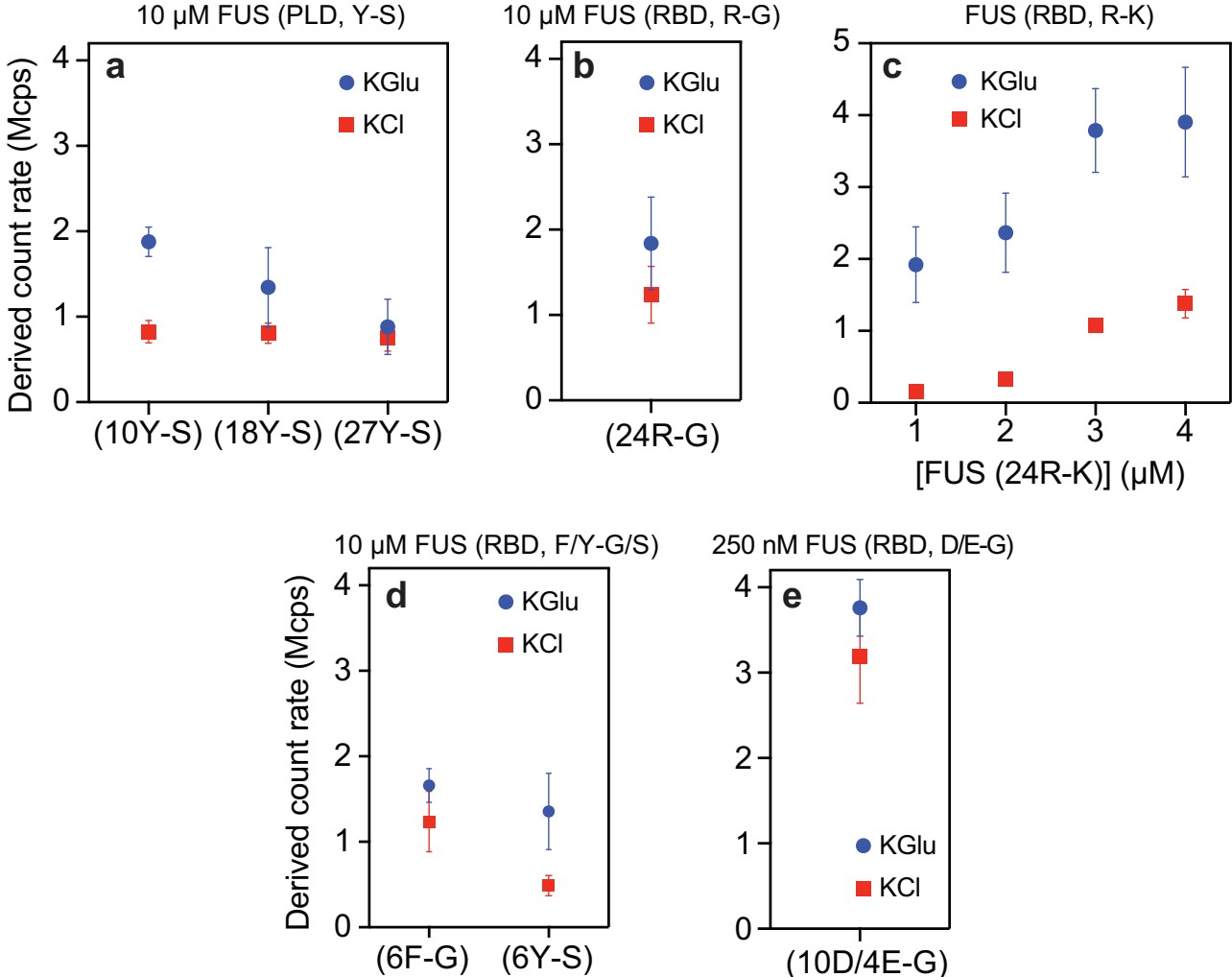

**Fig. 7 | Mutations modulate the extent of clustering in sub-saturated solutions, and this can be influenced by glutamate.** DLS data show the derived count rate of FUS(Y-S) (**a**), FUS(R-G) (**b**), FUS(R-K) (**c**), FUS(F/Y-G/S) (**d**), and FUS (10D/4E-G) (**e**) in different buffers. $n = 3$ independent samples were used for the measurements, and data are presented as mean values ± SD.

distribution in 100 mM KCl becomes bimodal, showing peaks at 2 ns and 4 ns. In the presence of 100 mM KGlu, the Nile Red lifetime distributions show one distinct peak with an average lifetime of 4 ns, which is reached at FUS-SNAP concentrations as low as 0.5 μM. The inference is that there is an increase in the number and size of clusters in KGlu compared to KCl (Fig. 9a).

To complement the analysis with Nile Red, we also used bis-ANS to probe the local environments within clusters that form in the presence of 100 mM KGlu versus 100 mM KCl (Fig. 9b, c). In both cases, in the presence of 2 μM bis-ANS, the fluorescence intensity increases with increasing protein concentration. The increase in intensity is higher in the presence of KGlu compared to KCl. As a control, when we increased the KCl concentration to 200 mM, it caused a decrease in the fluorescence intensity of bis-ANS with FUS-SNAP compared to 100 mM KCl buffer (Fig. 9d). This suggests that KCl inhibits the clustering of FUS-SNAP. To assess the apparent hydrophobicity of the clusters, the fluorescence intensity of the same concentration of bis-ANS was measured in methanol and ethanol (Supplementary Fig. 8b). The intensity of bis-ANS in the presence of FUS-SNAP clusters in KGlu buffer is comparable to that of bis-ANS in methanol. These findings suggest that clustering in sub-saturated solutions is enhanced in KGlu when compared to KCl. Stronger molecular associations increase the extent of clustering, as probed via the sizes of clusters, and the larger clusters are apparently more hydrophobic when compared to the surrounding solvent.

## Glutamate is preferentially excluded from sites on amino acids

To uncover the molecular basis for differences in cluster formation in KGlu versus KCl, we studied the interactions of the two salts with different amino acids using molecular dynamics (MD) simulations. These simulations use capped amino acids and explicit representations of solvent molecules and solution ions (see Methods). The simulation results were used to compute preferential interaction coefficients for the two salts for different amino acids (Glycine, Lysine, Arginine, Serine, Aspartic acid, Phenylalanine, Tyrosine, and Glutamine).

Preferential interaction coefficients quantify the excess numbers of cosolutes in the vicinity of the peptide when compared to the bulk solution. Using the local-bulk partitioning formalism[86] adapted for analysis of MD simulations[87] (Fig. 10a), preferential interaction coefficients were calculated using $\Gamma_{ion}(r) = N_{ion}(r) - N_{H_2O}(r)\left(\frac{N_{ion,bulk}}{N_{H_2O,bulk}}\right)$. Here, $N_{ion}(r)$ and $N_{H_2O}(r)$ quantify the numbers of ions and water molecules at a distance $r$ from the center-of-mass of the peptide, whereas $N_{ion,bulk}$ and $N_{H_2O,bulk}$ quantify the numbers of ions and water molecules in the bulk solution. The simulations show that, except for positively charged residues, the preferential interaction coefficients are lower for glutamate than chloride (Fig. 10b). This implies that glutamate is preferentially excluded from peptide sites. Preferential exclusion of glutamate will contribute to the increased association of macromolecules. For Arg and Lys, the anions localize around the amino acids

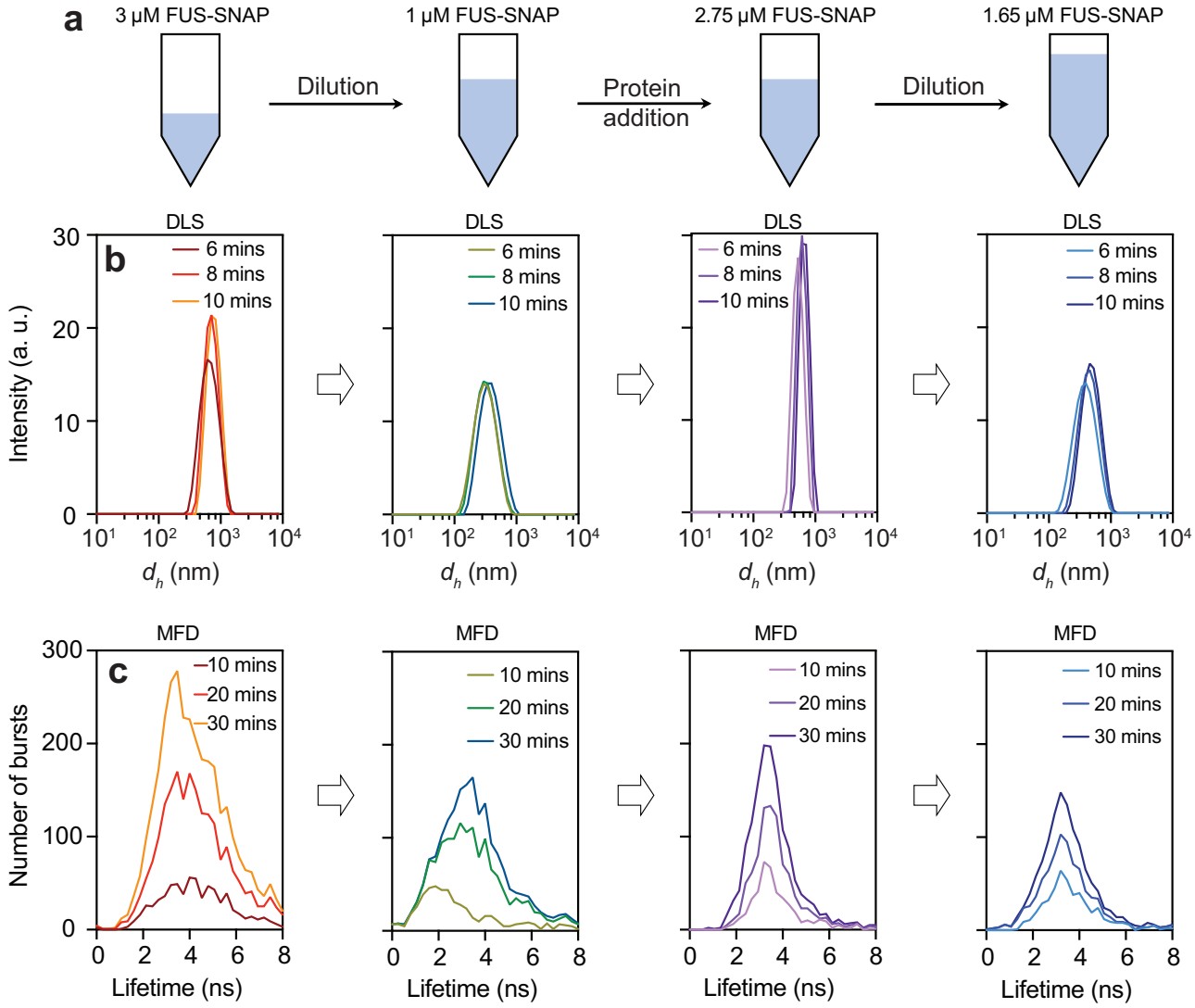

**Fig. 8 | FUS-SNAP clusters form reversibly in the 100 mM KGlu buffer. a** A schematic representation (drawn by the authors) of the reversibility experiments of FUS-SNAP clusters in KGlu buffer. **b** DLS data were collected at different time points for FUS-SNAP, showing the changes in intensity versus size profiles at 3 μM, 1 μM,

2.75 μM, and 1.65 μM. **c** The confocal point measurements with multiparameter fluorescence detection (MFD) display the number of bursts versus fluorescence lifetime of Nile Red at the same concentrations shown in (**b**), revealing shifts to lower lifetimes upon dilution.

to neutralize the charge. As a result, we do not observe substantial differences in preferential interaction coefficients of glutamate versus chloride around Arg and Lys, respectively.

Next, we computed preferential interaction coefficients for the whole salts using $\Gamma_{salt} = 0.5(\Gamma_{anion} + \Gamma_{cation} - |Z|)$, where $\Gamma_{anion}$ and $\Gamma_{cation}$ are the preferential interaction coefficients for the anion and cation respectively, and $Z$ is the net charge of the peptide (Fig. 10c). As with the values of $\Gamma_{anion}$, we find that the preferential interaction coefficients are smaller for KGlu when compared to KCl. This is true for all non-cationic amino acids. Radial distribution functions between the solution anions and different moieties on different peptides (Supplementary Figs. 9–12) show that the differences between glutamate and chloride originate from the differences in accumulation versus exclusion of anions around atoms of backbone and sidechain groups.

**Preferential exclusion of glutamate enhances associations of FUS**

We quantified the concentration of monomeric FUS-EGFP using a single photon counting confocal detection unit connected to a microfluidic flow cell. This enables the recording of intensity time

traces of clusters characterized by short bursts in intensity stemming from species passing through the confocal volume as well as the formation of a stable baseline intensity corresponding to the FUS-EGFP dilute phase concentration (Fig. 10d). Information regarding monomeric species can be extracted from the maximum intensity histogram as the dilute phase volume fraction far exceeds that of the dense phase (Fig. 10e). These measurements were performed for different KGlu concentrations at two different concentrations of FUS-EGFP to record how the concentrations of monomeric species change with glutamate concentrations (Fig. 10f). The background KCl concentration was kept constant at 40 mM. The concentrations of monomeric FUS-EGFP decreases upon increasing KGlu. The lowering of the concentration of monomeric FUS-EGFP is a signature of enhanced associations driven by preferential exclusion of glutamate from protein sites.

**Discussion**

In the mean-field formalism of Flory[32] and Huggins[33], the solubility parameter χ is proportional to the algebraic difference between the effective protein-solvent interactions and the arithmetic mean of protein-protein and protein-solvent interactions[5]. For associative

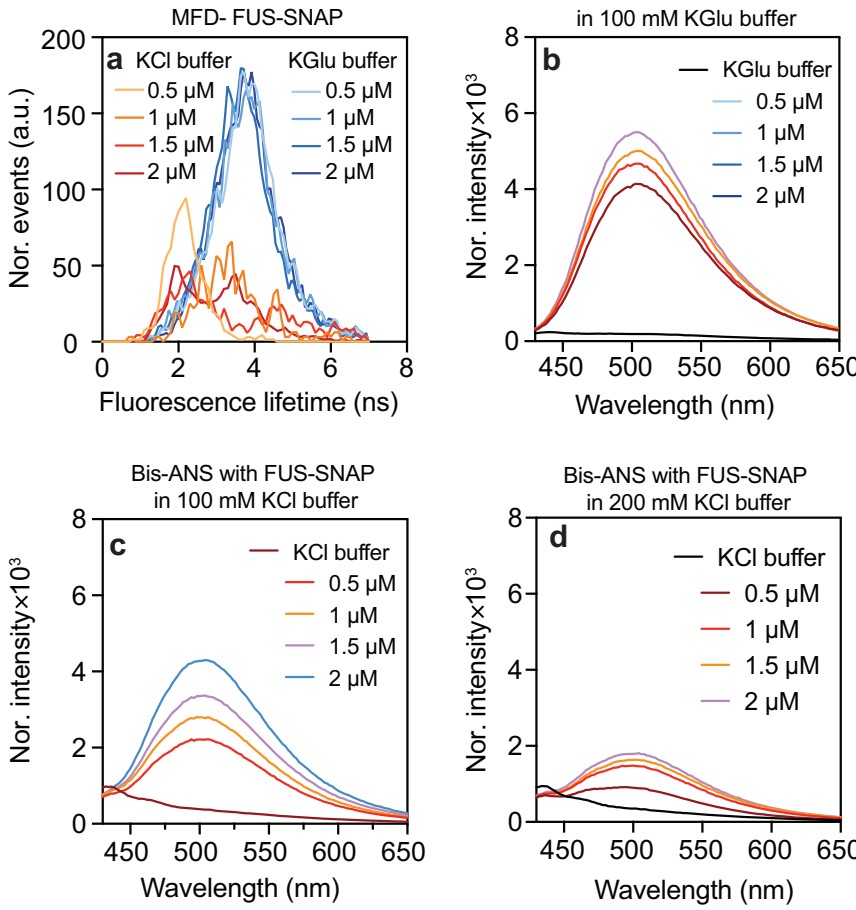

**Fig. 9 | FUS-SNAP clusters create distinct local environments. a** The fluorescence lifetimes of Nile Red with various concentrations of FUS-SNAP equilibrated for 30 min in KGlu and KCl buffer. The different concentrations of FUS-SNAP solutions and buffers are mixed with 2 μM bis-ANS in 100 mM KGlu buffer (**b**), 100 mM KCl buffer (**c**), and 200 mM KCl buffer (**d**). For bis-ANS studies (**b**–**d**), the mixture solutions were excited using a 355 nm laser, and the emission spectra were measured from 425 nm to 650 nm.

macromolecules, Tanaka introduced the concept of a renormalized χ to account for the effects of specific intermolecular associations, and the interplay with solvent-specific interactions[36]. The renormalized χ depends on macromolecular concentrations and it combines the contributions of specific sticker-stickers interactions, their mediation by solvent, and the effects of the interplay between solubility-determining macromolecule-solvent interactions and solvent-solvent interactions. In glutamate, our data show that associations of FET proteins are strengthened on the nanoscale. However, these do not translate to significant changes in $c_{sat}$. The implication is that for FET proteins, the renormalized χ is similar in chloride versus glutamate. This suggests that the enhancement of protein-protein interactions on nanoscales is counterbalanced by length-scale-dependent changes to macromolecule-solvent interactions and/or weakened solvent-solvent interactions in glutamate versus chloride. This would explain why cluster formation in sub-saturated solutions is enhanced, but $c_{sat}$ changes only minimally.

Our findings regarding the relative insensitivity of $c_{sat}$ to glutamate versus chloride differ from those of Kozlov et al. for bacterial SSBs[65], although even there $c_{sat}$ changes only by a factor 3-4. This suggests that the interplay of solvent-mediated specific associations and solubility-determining interactions are not generic across different systems. Instead, they are sequence- and architecture-specific. While FET proteins are flexible, linear associative polymers, the SSBs are protein-based exemplars of branched "hairy colloids"[88,89]. Taken together with the results of ref. 65, our work highlights the need for

comparing the effects of glutamate and other cellular metabolites on clustering and phase separation of different multivalent proteins defined by different sequence grammars and architectures.

Our comparative assessments of chloride versus glutamate were motivated by the fact that the latter is an exemplar of the types of anions that are present in cells. Yet, it is often assumed that physiologically relevant salt conditions correspond to 100–150 mM KCl or NaCl[90–94]. This assumption does not square with extant data for prokaryotic[58] or eukaryotic systems. For example, in the cytoplasms of glutamatergic neurons, the concentration of glutamate is in the 5–10 mM range[95]. In synaptic vesicles, glutamate concentrations can be as high as 100 mM[96]. Formulations for a "single-assay medium" intended to mimic the in vivo milieu of *Saccharomyces cerevisiae* include 300 mM K⁺, 245 mM glutamate, 50 mM phosphate, 20 mM Na⁺, 2 mM free Mg²⁺, all at a pH of 6.8[59]. Importantly, the mimicking medium does not include chloride. Therefore, RNA-binding proteins are unlikely to encounter chloride inside cells. Instead, glutamate or other metabolites including phosphates are likely to be the key anions, thus highlighting the biological relevance of findings reported in this work.

We used MD simulations to quantify preferential interaction coefficients for KCl and KGlu around amino acids with different side-chain chemistries. In line with the proposals of Record and coworkers[55,56,64,65], our simulations show that glutamate is preferentially excluded from backbone and sidechain amides, as well as other functional groups. However, there are key differences in the

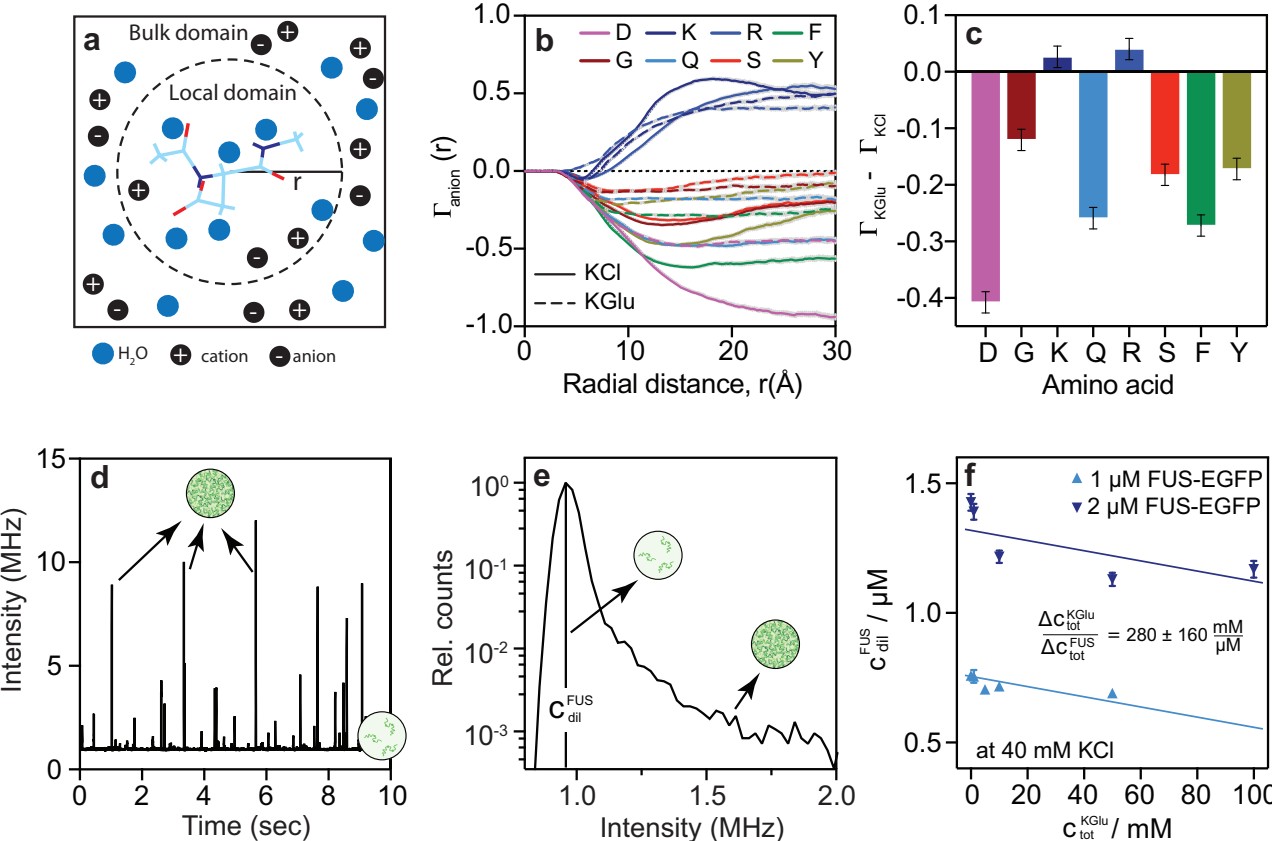

**Fig. 10 | Preferential exclusion of glutamate from peptide amide sites drives protein associations. a** Schematic for the calculation of preferential interaction coefficients as a function of distance $r$ from the center-of-mass of the peptide of interest. **b** Preferential interaction coefficients for different amino acids as a function radial distance $r$. The computations are of the anion-specific preferential interaction coefficients in 500 mM KCl and 500 mM KGlu, respectively. Error bars denote the standard error of the mean as obtained from 4 MD simulation replicates. **c** The difference in the preferential interaction coefficient for KGlu and KCl

$(\Gamma_{KGlu} - \Gamma_{KCl})$ at $r \approx 30\,\text{Å}$ for different amino acids. Error bars denote the standard error of the mean as obtained from 4 MD simulation replicates. **d** Representative intensity time trace of FUS-EGFP samples post induction of phase separation using a single photon counting detection unit. **e** The intensity histogram of the time trace is shown in (**d**). **f** The concentrations of monomeric FUS-EGFP decrease when KGlu concentrations are increased at a constant KCl concentration of 40 mM. $n = 3$ independent samples were used for all the measurements and data are presented as mean values ± SD.

atomic-level details that emerge from our simulations versus interpretations proposed by Record and colleagues[55,56,64,65].

Cheng et al. used vapor pressure osmometry (VPO) to measure the solubilities of model compounds in aqueous solvents with different types of solutes[55]. In their notation, water, the primary component is labeled 1, the model compound of interest is component 2, and the solute of interest, such as KGlu or KCl, is compound 3. The change in solubility, as gleaned from VPO measurements, leads to inferences regarding the sign and magnitude of the chemical potential $\mu_{23}$ for the preferential interaction of the model compound with the solute. A positive sign indicates preferential exclusion, whereas a negative sign implies preferential interactions. The measured $\mu_{23}$ values for fifteen different model compounds were decomposed using a global regression analysis based on a linear superposition model[55]. This model is written as $\mu_{23} = \sum \alpha_i A_i$. Here, the summation is over atoms of functional groups in the model compounds and $A_i$ is the solvent-accessible surface area of atom $i$ within the model compounds. The values of $A_i$ were computed using a specific probe radius and specific structures for each model compound. Based on the inferred values of $\alpha_i$, interactions of glutamate are proposed to be favorable for $sp^2$ nitrogen atoms and the nitrogen atoms of cationic residues. The converse was found to be true for chloride. Radial distribution functions from our simulations suggest that chloride interacts favorably with $sp^2$ nitrogen atoms when compared to glutamate (Supplementary Figs. 9–12). For $sp^2$ oxygen and backbone carbon atoms, ref. 55, inferred weaker interactions with

chloride when compared to glutamate, and our simulation results show similar trends (Supplementary Figs. 9–12).

Inferences from simulations were derived via a detailed accounting of the interplay of amino-acid, water, and solute interactions. Conversely, the VPO measurements data report one number for each model compound, and these are then dissected using an accessible surface area-based model combined with global regression analysis. The use of accessible surface area as a measure of solvation is problematic for small molecules and atomic-level dissections. Accessible surface area becomes a useful proxy only at large length scales[97,98]. This is because the concept of an interfacial tension does not apply on the atomic and molecular length scales. Instead, theory and simulation suggest that the hydration thermodynamics and forces require the inclusion of a volume term and dispersion interactions on atomic and molecular scales[99]. These are fully present in our simulations. Additionally, solvent-accessible surface areas are insensitive to changes in conformation and the local concentrations of functional groups for linear, flexible systems[100]. Hence, while the use of solvent-accessible surface area is widely prevalent in the protein folding literature, and has been justified by elegant connections to Kirkwood-Buff integrals[101], their use for dissecting atomic-level interactions of disordered proteins is problematic. The preferential interaction coefficients we computed were directly gleaned from pair distribution functions, in accord with the Kirkwood-Buff formalism[102]. At this juncture, we lean on consistencies of interpretations from the work of ref. 55, and the

simulations. Both studies suggest that the central differences between chloride and glutamate derive from the latter being preferentially excluded from protein sites.

Overall, our work highlights how components of cellular milieus contribute to amplifying the distinct contributions from pre-percolation clusters and phase separation, respectively[5]. Our results imply that even if the endogenous levels of FET proteins are below $c_{sat}$, we should expect these and other proteins like them to form heterogeneous distributions of clusters in cellular milieus. The abundance of clusters and the size distributions of clusters will be governed by the expression levels in cells. These clusters, which are different from micron-scale condensates, may be of direct functional relevance in vivo[54,103–106].

## Methods
### Protein purification
All proteins were expressed in SF9 insect cells cultured at 27 °C in suspension in glass culture flasks in ESF921 serum-free medium. Baculovirus for expression of target genes under the control of the polyhedrin promoter was generated using the FlexiBAC system[107]. In a typical experiment, 1 L SF9 insect cells at a density of 1 million cells per mL were infected with 5 mL P2 virus and harvested 72 h post-infection. Cells were centrifuged at 300 RCF for 15 min at 4 °C. The pellet was resuspended in 30 mL ice-cold lysis buffer (50 mM Tris.HCl pH 7.4, 1 M KCl, and 5% Glycerol) supplemented with EDTA-free Protease Inhibitor Cocktail Tablets (1 tablet/ 100 mL). The cells were lysed by 5 min of sonication on ice at output level 35, with 50% duty cycle. Unbroken cells and debris were removed by centrifugation at 39,800 RCF for 30 min at 4 °C. All subsequent steps of the purification were performed at room temperature. The supernatant was passed over a pre-packed 5 ml Ni-NTA agarose column (Protino, Macherey-Nagel) using a peristaltic pump. The column was washed with 10 column volumes (CV) of lysis buffer supplemented with 10 mM imidazole. The target protein was eluted with 5 CV NTA elution buffers (50 mM Tris.HCl pH 7.4, 1 M KCl, 5% Glycerol, and 300 mM Imidazole). Pooled peak fractions were incubated with 10 ml amylose resin for 10 min, and the column was subsequently drained by gravity flow. The resin was washed with 10 CV lysis buffer, and the target protein was eluted with 5 CV MBP elution buffer (50 mM Tris.HCl pH 7.4, 1 M KCl, 5% Glycerol, and 30 mM Maltose). Protein concentration was continuously monitored using a Bradford assay, and the purity of each fraction was assessed by SDS-PAGE.

To cleave off the N-terminal His-MBP tag, 3C precision protease was added to the eluted protein at a 1:100 molar ratio. The mixture was incubated at room temperature for 4 hrs and subsequently submitted to size-exclusion chromatography using ÄKTA (GE Healthcare) with Superdex 200 10/300 increase column equilibrated with storage buffer (50 mM Tris.HCl pH 7.4, 500 mM KCl, 5% Glycerol, and 1 mM DTT). C-terminal SNAP-tagged protein peak fractions were pooled, concentrated, and used for the experiments.

To obtain untagged protein, TEV protease was added at a 1:50 molar ratio and incubated at room temperature for 6 h. The untagged protein was purified using gel filtration chromatography (ÄKTA with Superdex-200 increase 10/300 column) and equilibrated with storage buffer. Peak fractions were pooled and concentrated using a 30 kDa molecular weight cut-off (MWCO) at 3000 RCF at room temperature. Protein concentration was determined using a NanoDrop ND-1000 spectrophotometer (Thermo Scientific). The 260/280 ratio of all purified proteins was measured to be between 0.52 to 0.56. Peak fractions were pooled and used immediately for the DLS or NTA experiments.

SNAP-tagged proteins were snap-frozen in liquid nitrogen and stored at −80 °C. Immediately prior to experiments, frozen proteins were thawed, and the tag was cleaved by TEV digestion and subsequent size-exclusion chromatography. Peak fractions were pooled, concentrated, and used for the experiments. This protocol allows for data reproducibility.

### Determination of saturation concentrations
We prepared the samples at the indicated protein concentration in 100 mM KCl and KGlu buffers, and 30 min after sample preparation, the samples were spun down at 20,000 RCF on a benchtop centrifuge unit and measured the concentration in the clarified supernatant by using a Bradford assay[108].

### Microscopy measurements of phase separation
For droplet formation assays, proteins were diluted into various concentrations in the corresponding 100 mM KCl buffer, 100 mM KGlu buffer, and KCl-control buffers in a total solution volume of 30 μL. The samples were added into the 384-well non-binding microplates (Greiner bio-one). The images were taken after various time points, starting from 1 h to 4 h. Images were taken using an IX71/IX81 inverted Spinning Disc Microscope with an Andor Neo sCMOS/Andor Clara CCD camera and a UPlanSApo 60x oil-immersion objective (Olympus).

### Reproducibility
Three independent microscopy measurements ($n = 3$) were performed. For each condition and time point, ten different micrographs were acquired from various parts of the same sample well. The represented micrograph data displays a similar outcome to that of several micrographs.

### DLS Measurements
DLS measurements were performed using the Zetasizer Nano ZSP Malvern instrument (measurement range of 0.4 nm to 10 μm). The Nano ZSP instrument incorporates noninvasive backscattering technology. This enables the measurement of time-dependent fluctuations of the intensity of scattered light as scatterers undergo Brownian motion. The analysis of these intensity fluctuations enables the determination of the diffusion coefficients of particles, which are converted into a size distribution using the Stokes-Einstein equation[109]. A 632.8 nm laser illuminated the sample solutions, and the intensity of light scattered at an angle of 173° was measured using a photodiode.

In DLS, the autocorrelation function of the scattered light is used to extract the size distribution of the dissolved particles. The first order electric field correlation function of laser light scattered by a monomodal or monodisperse population of macromolecules can be written as a single exponential shown in Eq. (1):

$$G(\tau) = 1 + b\,exp\left(-2D_t q^2 \tau\right); \tag{1}$$

Here, $b$ is a constant that is determined by the optics and geometry of the instrument, $D_t$ is the translational diffusion coefficient of the particles, and $\tau$ is the characteristic decay time. The scattering vector $q$ is given by Eq. (2):

$$|q| = \frac{4\pi n_0}{\lambda_0}\sin\left(\frac{\theta}{2}\right); \tag{2}$$

Here, $n_0$ is the refractive index of the solvent, $\lambda_0$ is the wavelength, and $\theta$ is the scattering angle. For populations composed of a single type of scatterer, the distribution function of decay rates can be derived from a simple fit of the experimental estimates of the logarithm of the correlation function in Eq. 1 to a polynomial. These methods, which apply to monomodal distributions of sizes of scatterers, can be used to extract the translational diffusion coefficient, from which one can estimate the hydrodynamic radius $R_h$ of the scatterers. For this, one uses the Stokes-Einstein relation in Eq. (3):

$$D_t = \left(\frac{k_B T}{6\pi \eta R_h}\right); \tag{3}$$

Here, $k_B$ is the Boltzmann constant ($1.381 \times 10^{-23}$ J/K) and η is the absolute (or dynamic) viscosity of the solvent. In this work, we used the hydrodynamic diameter $d_h$ (i.e., $d_h = 2R_h$) as the preferred way to quantify particle sizes.

For the measurements, all solutions were filtered using 0.2 µm membranes (Millex®-GS units) purchased from Millipore™. All experiments were conducted with the following settings on the Malvern instrument: Material−protein; Dispersant−buffers; Mark-Houwnik parameters; Temperature: 25 °C with equilibration time−120 s, Measurement angle: 173°. Each spectrum represents the average of 12 scans each of 10 s in duration. All proteins were freshly purified and used after chromatography purification with a standard stock solution buffer. The samples were prepared by adding freshly prepared stock proteins followed by dilution buffer and mixed thoroughly by pipetting four to six times. The samples were equilibrated for 2 min at 25 °C, and the data were recorded in 2-min intervals.

### Nanoparticle tracking analysis (NTA)

NTA was performed using NS300 from Malvern instruments (measurement range of 20 nm to 1 µm). The system was accompanied by a NanoSight syringe pump to inject the samples for the experiments. NTA measurements utilize the properties of light scattering and Brownian motion to quantify the size distributions and concentrations of particles in liquid suspension. A laser beam (488 nm) was passed through the sample chamber, and the particles in suspension were visualized using a 20x magnification microscope. The video file of particles moving under Brownian motion was captured using a camera mounted on the microscope that operates at 30 frames per second. The software tracks particles individually and uses the Stokes-Einstein equation to resolve particles based on their hydrodynamic diameters.

All proteins were freshly purified and used after chromatographic purification with a standard stock solution buffer. All buffers were filtered through a 0.22 µm polyvinylidene fluoride membrane filter (Merck, Germany). All protein stock solutions were centrifuged at 20,000 RCF for 5 min at room temperature before measurements. The samples were prepared by adding freshly prepared, centrifuged stock proteins followed by dilution buffer and mixed thoroughly by pipetting 4−6 times. The samples were equilibrated for 2 min at 25 °C, and the data were recorded 6 min after sample preparation and equilibration.

### Multiparameter fluorescence detection (MFD) measurements and analysis

**Confocal point measurements.** All confocal point measurements were conducted on a confocal fluorescence microscope (Olympus IX71, Hamburg, Germany) using a supercontinuum laser (SuperK Extreme with SuperK VARIA tunable filter, NKT Photonics, Birkerød, Denmark) at 514 nm ± 1.5 nm and 39 MHz. Laser light was directed into a 60x water immersion objective (NA = 1.2) by a dichroic beam splitter and focused into the sample close to the diffraction limit volume. The emitted light was collected by the same objective and separated into two polarizations (parallel and perpendicular) relative to the excitation beam. The fluorescence signal was further divided into two spectral ranges by beam splitters (BS 560, AHF, Tübingen, Germany). Additionally, Bandpass filters for Nile Red fluorescence (HC 607/70) were placed in front of the detectors. The signals from single photon counting detectors (SPCM-AQRH-14 TR Excelitas, Wiesbaden, Germany) were recorded photon-by-photon with picosecond accuracy (HydraHarp400, PicoQuant, Berlin, Germany) and analyzed using custom software (LabVIEW based, see https://www.mpc.hhu.de/software/mfd-fcs-and-mfis). The temperature in the laboratory during all titration steps was 20 ± 1 °C.

FUS-SNAP or untagged FUS was titrated into 15 nM Nile Red solution at standard buffer conditions (20 mM Tris.HCl, pH 7.6) with either 100 mM KCl or 100 mM KGlu. The KGlu buffer contains 10 mM residual KCl from the FUS stock solution. Both dye and salt concentrations are kept constant during the titration and back-dilution using the respective Nile Red-containing buffer solutions.

**Burst analysis.** Using in-house LabVIEW based software, bursts were selected via an intensity threshold using 2σ criteria out of the mean background with a photon minimum number of 10 photons per burst. The decays for each burst were then processed and fitted using a mono-exponential model yielding fluorescence-weighted average lifetime considering correction factors for background signal, polarization (g-factor), and scattering effects. Especially for FUS titrations in the nM to µM regime, an additional photon maximum of 3000 was applied due to computational limitations.

**Fluorescence intensity distribution analysis (FIDA).** We performed FIDA for all experiments according to established protocols[110].

**Single-molecule FRET (smFRET) measurements.** smFRET experiments were conducted using a confocal epi-illuminated setup based on an Olympus IX71 microscope deploying pulsed interleaved excitation (PIE) where the donor and acceptor fluorophore are sequentially excited by fast alternating laser pulses thus allowing the computation of the stoichiometry S (donor-acceptor-ratio). Excitation is attained using 485 nm (50 µW) and 640 nm (10 µW) pulsed diode lasers (LDH-D-C 485 and LDH-P-C-635B, PicoQuant Berlin, Germany) operated at 32 MHz and focused by a 60×/1.2 NA water immersion objective (UPLAPO 60x, Olympus, Hamburg, Germany) into the sample. We used the excitation beam splitter FF500/646 (Semrock, USA), a polarizing beam splitter cube (VISHT11, Gsänger), and dichroic detection beam splitters (595 LPXR, AHF, Tübingen, Germany) to separate fluorescence from laser excitation and split it into its parallel and perpendicular spectral components. The four detection channels, corresponding to color and polarization, were split further by 50/50 beam splitters to obtain dead time-free species cross-correlation curves, yielding a total of eight fluorescence detection channels (green channels: τ-SPAD-100, PicoQuant; red channels: SPCM-AQR-14, Excelitas, Wiesbaden, Germany). To block out Raman scattering, green (HQ 530/43 nm for FUS-SNAP-Alexa488) and red (HQ 720/150 nm for FUS-SNAP-Alexa647) bandpass filters (AHF, Germany) were put in front of the corresponding detectors. The detector outputs were recorded by a TCSPC module (HydraHarp400, PicoQuant). Measurement times for single-molecule detection (SMD) experiments were about 10 h each.

The equations for the static and dynamic FRET lines, including the weighted lifetimes for both species, the correction parameters, and all 2D-FRET efficiency plots, are shown in Eqs. (4) and (5):

$$E_{static} = 1 - \frac{0.0065\tau_{(D)A}^{4} + \left(-0.0927\tau_{(D)A}^{3}\right) + 0.4244\tau_{(D)A}^{2} + 0.3738\tau_{(D)A} - 0.0215}{3.9000} \tag{4}$$

$$E_{dynamic} = 1 - \frac{1.70003.7000}{3.9000(1.7000 + 3.7000 - (1.3337\tau_{(D)A} - 1.2360))} \tag{5}$$

**Fluorescence correlation spectroscopy (FCS) curve fitting.** Donor autocorrelation curves of FUS-SNAP-AF488 in smFRET measurements were fitted for correlation times $t_c$ from $10^{-4}$ to $10^2$ ms using a 3D-Gaussian model with two diffusion terms for two selections: (i) monomer cut: $t_{d,dye}$, translational diffusion time of free dye impurities, which are global for the three data sets of the buffers as well as the reference, free Rhodamine 110, and $t_{d,mo}$, translational diffusion time of the monomer components are shown in

Eqs. (6a) and (6b):

$$G(t_c) = G_0$$

$$+ \frac{1}{N} \cdot \left( \frac{|f|}{1 + \frac{t_c}{|t_{d,dye}|}} \cdot \frac{1}{\sqrt{\left(1 + \frac{t_c}{\left(\frac{z_{0,1}}{\omega_{0,1}}\right)^2 \cdot |t_{d,dye}|}\right)}} + |1 - |f|| \cdot \frac{1}{1 + \frac{t_c}{|t_{d,mo}|}} \cdot \frac{1}{\sqrt{\left(1 + \frac{t_c}{\left(\frac{z_{0,2}}{\omega_{0,2}}\right)^2 \cdot |t_{d,mo}|}\right)}} \right)$$

$$\cdot \left( 1 - |A| + |A| \cdot e^{-\frac{t_c}{|t_A|}} \right) \tag{6a}$$

(ii) cluster cut: $t_{d,mo}$, translational diffusion time of the monomer component, which are global for both data sets of the buffers, and $t_{d,cl}$, average translational diffusion time of the cluster components:

$$G(t_c) = G_0$$

$$+ \frac{1}{N} \cdot \left( \frac{|Rf|}{1 + \frac{t_c}{|t_{d,mo}|}} \cdot \frac{1}{\sqrt{\left(1 + \frac{t_c}{\left(\frac{z_{0,1}}{\omega_{0,1}}\right)^2 \cdot |t_{d,mo}|}\right)}} + |1 - |R|| \cdot \frac{1}{1 + \frac{t_c}{|t_{d,cl}|}} \cdot \frac{1}{\sqrt{\left(1 + \frac{t_c}{\left(\frac{z_{0,2}}{\omega_{0,2}}\right)^2 \cdot |t_{d,cl}|}\right)}} \right)$$

$$\cdot \left( 1 - |A| + |A| \cdot e^{-\frac{t_c}{|t_A|}} \right) \tag{6b}$$

The descriptions of all other parameters in Equation 6 a and b and the fit results are compiled Supplementary Table 2 and Supplementary Table 3.

Using the translational diffusion times of the single-molecule FUS measurements together with the Rhodamine 110 reference ($D_{ref}$ = 4.3E6 cm$^2$/s) a translational diffusion coefficient for both buffers is calculated as in Eq. (7):

$$D_{buffer} = \frac{D_{ref} \cdot t_{d1,global}}{t_{d2,buffer}} \tag{7}$$

The hydrodynamic radius $R_h$ is computed by the Stokes-Einstein equation at a temperature T = 293.15 K as in Eq. (8).

$$R_{h,buffer} = \frac{kT}{6\pi\eta D_{buffer}} \tag{8}$$

**Maximum entropy method (MEM)[78].** The fitting model above (eq.6) assumes two distinct diffusion times. We have used a MEM to investigate whether more complicated distribution times of diffusion times better fit the experimental data. The diffusional factor of a correlation ($G_d$) was presented as the weighted sum of diffusional terms for a fixed set of diffusion times $t_{d,i}$ as in Eq. (9):

$$G_d(t_c) = \sum_{i=1}^{N} p_i G_d(t_c; t_{d,i}),$$
$$G_d(t_c; t_{d,i}) = \frac{1}{1 + \frac{t_c}{t_{di}}} \cdot \frac{1}{\sqrt{\left(1 + \frac{t_c}{\left(\frac{z_0}{\omega_0}\right)^2 \cdot t_{di}}\right)}} \tag{9}$$

The vector of weighting factors $p = (p_1, \ldots, p_N)$ minimizing the regularized functional shown in Eq. (10)

$$Q(p) = \chi^2(p) - \nu S(p), \tag{10}$$

was found by the method of a quadratic programming following ref. 79. In Eq. (10) the term $\chi^2(p)$ is the least-squares goodness-of-fit functional expressed in the quadratic form shown in Eq. (11):

$$\chi^2(p) = \chi_0^2 + q \cdot p + p \cdot H \cdot p, \tag{11}$$

Here, the vector $q$ and matrix $H$ are defined by experimental data. The factor $S(p)$ in the second term is the Kullback-Leibler relative entropy shown in Eq. (12). Here, the prior distribution is $m = (m_1, \ldots, m_N)$:

$$S(p) = -\sum_i \left( p_i \ln\left(\frac{p_i}{m_i}\right) \right) \tag{12}$$

We used a uniform priori $m$. The factor $\nu$ is the regularization parameter chosen such way that the optimized vector of amplitudes $p$ provide maximum entropy $S(p)$ for reasonably low values of the goodness-of-fit parameter $S(p)$. The low value of $\nu$ leads to a solution (low entropy) with multiple separated sharp peaks, sometimes false ones. The increasing of regularization makes solution smother (increase entropy). The lowest value of an entropy (zero) is achieved for uniform $p_i$ in accordance with our choice of $m$.

**Microfluidic confocal spectroscopy (MCS).** Microfluidic devices[111], were first fabricated as SU-8 molds (MicroChem) through standard photolithographic processes and then produced as poly-dimethylsiloxane (PDMS) slabs, which were bonded onto thin glass coverslips[112]. The devices were operated by placing gel-loading tips filled with buffer and protein samples in their corresponding inlet ports and pulling solution through the devices in withdraw mode at a flow rate of 150 μL/h using automated syringe pumps (neMESYS, Cetoni).

The FUS-EGFP was stored in 20 mM Tris.HCl pH 7.4, 500 mM KCl was diluted with buffers of 20 mM Tris.Glu to the indicated protein and KGlu concentrations as stated. During the experiment, the sample was placed into the sample inlet of the device, and the corresponding buffer contained the same concentration of KGlu in the buffer inlet. The co-flowing buffer was supplemented with 0.05% Tween-20 to prevent surface sticking of the protein to PDMS and glass surfaces.

Experiments were conducted by scanning the confocal spot of a custom-built confocal microscope through the central four channels of the microfluidic device[52]. Briefly, the setup is equipped with a 488-nm laser line (Cobolt 06-MLD) for excitation of EGFP fluorophores and a single-photon counting avalanche photodiode (SPCM-14, Perki-nElmer) for subsequent detection of emitted fluorescence photons. Further details of the optical unit have been described previously. During the scanning of the device, 200 evenly spaced locations within the central four channels of the device were surveyed and detected for 4 s. Clusters were classified as peaks that exceeded 5 standard deviations above the mean fluorescence intensity of each trace. These peaks were quantified according to location against the mean signal of each trace. The average number of clusters $\bar{n}_{clusters}$ was then quantified by averaging each of the four groups of peaks corresponding to the four central channels. This was used in the calculation of cluster concentration according to the following equation:

$$F_{total} = \left(\frac{\bar{n}_{clusters}}{t}\right)\left(\frac{4hd_{step}}{\pi zw}\right); \tag{12}$$

Here, $t$ is the time each trace was collected for (4 s), $h$ is the height of the microfluidic channel (28 μm), $d_{step}$ is the width of each step (5.64 μm), and $z$ and $w$ were the height and width of the confocal spot (3 μm and 0.4 μm, respectively). From Eq. (12), which yields the flux of clusters $F_{total}$, the concentration of clusters could be determined according to Eq. (13), with $Q_{sample}$ being the flow rate of the sample (15 μL/h) and $N_A$ being the Avogadro constant[74]:

$$c_{cluster} = \left(\frac{F_{total}}{N_A Q_{sample}}\right); \tag{13}$$

**Nano differential scanning fluorimetry (nanoDSF).** Thermal unfolding of FUS-SNAP was performed using nanoDSF with a Prometheus NT.48 (NanoTemper Technologies, München, Germany) instrument.

Protein samples were prepared in buffers and centrifuged shortly (5 min, 10,000 RCF) to remove protein aggregates. Samples were loaded into high-sensitivity glass capillaries (Cat#PR-C006, NanoTemper Technologies, München, Germany) and exposed at a linear thermal ramp from 20 °C to 95 °C by thermal ramping rate of 1 °C/min. Intrinsic protein fluorescence emission was collected at 330 and 350 nm with a dual-UV detector over a temperature gradient. The fluorescence intensity ratio (350/330) was plotted against the temperature, and the inflection point of the transition was derived from the maximum of the first derivative for each measurement using Therm-Control Software (NanoTemper Technologies, München, Germany). All experiments were carried out in triplicate; mean and standard deviation were calculated for all three measurements.

**Measurements of bis-ANS fluorescence.** Different concentrations of FUS-SNAP protein were mixed with 2 μM bis-ANS solution in 100 mM KCl and KGlu buffers. Then, the solutions were mixed and loaded 100 μL in a 96-well plate (microplate, PS, half area, μClear, Med. binding, Black, Greiner Bio-one). The spectra were recorded from 425 nm to 650 nm (10 nm bandwidth) with the TECAN plate reader using an excitation wavelength of 355 ± 5 nm. For control, we used only 2 μM bis-ANS with the same buffer conditions, ethanol, and methanol.

**Molecular dynamics (MD) simulations.** We used the Charmm36[113] forcefield to perform MD simulations using the GROMACS 2021 package[114,115]. Simulations were performed with explicit representations of solvent molecules using the TIP3P[116] water model. The simulation setup was as follows: we used capped amino acids of the form Ace-Xaa-Nme, where Ace refers to N-acetyl, Nme refers to N′-methylamide, and Xaa is the residue of interest, which is one of Gly, Asp, Glu, Arg, Lys, or Gln. Then, we solvated the peptide by using the default "scale" factor of 0.57 in a cubic simulation box with box size $7 \times 7 \times 7 nm^3$. Ions were added to the simulation box by replacing water molecules to neutralize the charge on amino acid (if any) and to obtain the salt concentration of 500 mg / mL. On average, we included 11,007 water molecules for the simulations where KCl is used and 10,333 water molecules for simulations with KGlu.

We performed energy minimization using steepest descent followed by 100 ns equilibration at 298 K and 1 bar. We then performed additional simulations, each 400 ns long to obtain production runs which were later used to obtain radial distribution functions and preferential interaction coefficients. Periodic boundary conditions were employed in all three directions. The V-rescale thermostat[117] was used to maintain the temperature at 298 K with a coupling time constant of 0.1 ps. The pressure was maintained by using the Parinello-Rahman[118] method with a time constant of 2.0 ps. Long-range electrostatic interactions were handled using the smooth Particle-mesh Ewald (SPME) algorithm[119]. We used the cutoff for short-ranged Lennard-Jones potential to be 1.1 nm while the cutoff for the short-ranged electrostatic potential was 1.2 nm. The LINCS algorithm[120] was used to constrain covalent bonds involving hydrogen atoms.

**Calculation of preferential interaction coefficients.** These coefficients are measures of the amount of cosolute in the local domain of the peptide compared to the solvent[87]. We determined the number of ions ($N_{ions(r)}$) and the number of water molecules ($N_{H_2O}(r)$) as a function of distance $r$ from the center of mass of the peptide, by using distance bins of size 0.2 Å. We then found the ratio of bulk density of ions to water molecules by calculating the ratio $\frac{N_{ion,bulk}}{N_{H_2O,bulk}} = \frac{N_{ion}(r \geq 2.5nm \text{ and } r \leq 3.38nm)}{N_{H_2O}(r \geq 2.5nm \text{ and } r \leq 3.38nm)}$ We used 400 ns of production runs across 4 replicates to obtain preferential interaction coefficients as a function of distance $r$ from the center of mass of the peptide.

## Data availability

Source data are provided as a Source Data file via the GitHub repository of the Pappu lab (https://github.com/Pappulab/Glutamate_vs_Chloride_Clustering/). The MD simulations data generated in this study have been deposited in the Zenodo database under accession code https://doi.org/10.5281/zenodo.10593297. Source data is also available from the corresponding author upon request.

## Code availability

All custom-made code for the analyses can be found on the GitHub repository of the Pappu lab (https://github.com/Pappulab/Glutamate_vs_Chloride_Clustering/). MD simulations were performed using the GROMACS 2021 package[121], https://manual.gromacs.org/2021/index.html.

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

## Acknowledgements

We are grateful to Andrei Pozniakovsky for the DNA constructs of all proteins, to Régis Lemaitre and Barbara Borgonovo for technical support with protein expression, purification, and characterization at MPI-CBG, and to Ralf Kühnemuth and Oleg Opanasyuk for technical assistance. We thank Eric Geertsma, Timothy Lohman, and Min Kyung Shinn for helpful discussions. This work was funded by a direct grant from the Max Planck Society (to A.A.H.), a grant from the NOMIS foundation (to A.A.H.), the Wellcome trust (209194/Z/17/Z to A.A.H.), the Deutsche Forschungsgemeinschaft (DFG) under Germany's Excellence Strategy—EXC-2068—390729961- Cluster of Excellence Physics of Life of TU Dresden (to A.A.H.), the European Research Council through the ERC grant PhysProt (to T.P.J.K., agreement no. 337969), the Wellcome Trust and the Frances and Augustus Newman foundation (to T.P.J.K.), SPP2191 from the Deutsche Forschungsgemeinschaft (to C.A.M.S. and R.V.P.), the US National Institutes of Health (R01NS121114 to R.V.P.), the US National Science Foundation (MCB-2227268 to R.V.P.), and the St. Jude Children's Research Hospital collaborative research consortium on the Biology and Biophysics of RNP Granules (to R.V.P.).

## Author contributions

M.K., A.A.H. and R.V.P. designed the project. M.K. discovered the effects of glutamate on the clustering of FET family proteins in sub-saturated solutions. M.K. prepared the samples and performed the Bradford assays, DLS, NTA, bis-ANS, nanoDSF, and microscopy measurements. A.R.K. assisted M.K. with microscopy measurements. M.K. and R.V.P. analyzed the spectroscopy data. C.A.M.S. and L.T.V. designed, performed, and analyzed the smFRET, FCS, and MFD, measurements using samples provided by M.K. S.F. assisted with analysis of the data collected by L.T.V. G.C. performed the MD simulations and analyzed the results with inputs from R.V.P. T.J.W., H.A. and T.P.J.K. designed, performed, and analyzed the MCS measurements using samples provided by M.K. M.K., L.T.V., G.C., F.D. and R.V.P. designed the structure of the manuscript and the layout of the figures. M.K., L.T.V., and G.C. put together all the figures with inputs from C.A.M.S. and R.V.P. R.V.P. wrote the manuscript, building on an initial draft provided by M.K. Significant inputs in editing and revising the manuscript were provided by M.K., L.T.V., G.C., and C.A.M.S. F.D., A.A.H., C.A.M.S. and R.V.P. provided critical intellectual inputs. T.P.J.K., A.A.H., C.A.M.S. and R.V.P. secured funding for the work. All authors read and edited the manuscript.

## Competing interests

R.V.P. is a member of the scientific advisory board and shareholder of Dewpoint Therapeutics Inc. A.A.H. is the founder, board member, and shareholder of Dewpoint Therapeutics. These affiliations did not have any bearing on the work reported here. The remaining authors declare no competing interests.
