## [Peer Review File · Nature Communications]

Glutamate unmask differences in driving forces for clustering versus phase separation of FET proteinsREVIEWER COMMENTS

Reviewer #1 (Remarks to the Author):

In this manuscript, the authors investigated the effects of glutamate versus chloride on both macrophase separation and clustering in sub-saturated solutions of FUS and specific FET family proteins. It proved that the macrophase separation and clustering of the proteins are two distinct and separable processes, by using two different salts (glutamate and chloride) as the background. This manuscript can be improved on following aspects.

1. The conclusion of this paper is similar to the previous work of the authors (reference 7) but different methods were used. Therefore, the authors need to state the scientific novelty and its difference with the previous work.
2. Introduction. Please give explanation to the fact that ATP and 1,6-hexanediol dissolve condensates while do not affect the clustering.
3. Page 9. The authors used two fluorescent probes, Nile red and bis-ANS. As Nile red shows increased quantum yields in nonpolar environments, what is the property of bis-ANS in response to its environment?

Reviewer #2 (Remarks to the Author):

Kar and coworkers present a combined experimental/computational analysis of the impact of glutamate anions, major constituents of the cytosol, on the formation of biomolecular condensates of protein FUS and other members of the FET protein family. As the main theme of the study, they compare and contrast the impact of glutamate with that of chloride ions within a framework in which percolation is coupled with phase transition in condensate formation. The question of the mechanistic foundation of biomolecular condensate formation is an important one and the authors provide several relevant and thought-provoking results in this regard. However, there is a number of key issues that need to be addressed adequately before the suitability of the manuscript for publication in Nature Communications can be assessed.

Major comments

1. A difficulty with reading the article comes from the fact that the authors do not provide a clear description of the expected quantitative, experimentally observable differences between phase separation and percolation processes. What does one expect if the observed clusters (meso- or macro-) are formed by separation or percolation? Does one expect a difference in cluster size distribution for the same concentration point? Intuitively, phase separation vs. percolation should behave quite differently since in the first case it is an all-or-nothing transition (no clusters observed until threshold concentration) and in the latter case a continuum of clusters should appear at concentrations below the percolation threshold.
2. The experimental methods used in the study rely mostly on monitoring particle diffusion (e.g. DLS, FCS, NTA) and, as such, they report on the apparent sizes, but not the masses of cluster i.e. the number of proteins in them. It is important that the effects are clearly demonstrated for the mass distribution of clusters. For example, can the derived count rates in DLS data be converted to SLS intensities and be related to mass (Figure 2 g-i, Figure S3 g-h)? This is important as the observed changes in the apparent hydrodynamic size can have many different explanations, e.g. protein can become swollen in one buffer and compact in another and the same could translate to their respective oligomers. In other words, two clusters can contain the same number of proteins, while being of a different hydrodynamic diameter. Conversely, they can have the same diameter, while containing a very different number of proteins.
2. In light of this, the authors should directly assess the conformational properties of single protein

molecules in different environments.

3. Importantly, in the absence of a precise estimation of the size of the clusters in terms of mass or number of proteins, any thermodynamic arguments or estimates related to the free energy of transfer of FUS molecules from the dispersed phase into mesoscale clusters in glutamate versus chloride cannot be made. For example, on L172-173, the authors estimate that this difference is 0.5-0.75 kcal/mol, but this must come from a tacit assumption that the abundance of mesoscale clusters is a direct measure of the amount of protein molecules contained in them, which need not be the case. As already mentioned, the authors should measure the sizes of clusters in terms of the mass of proteins they contain or, at least, clearly state and critically discuss the assumptions behind connecting cluster sizes with protein mass. In general, the authors should from the start precisely define what they mean by "size of clusters" throughout. This is not just a semantic point, but rather concerns the very essence of the arguments made.

4. Record and coworkers (PMID: 27806267, 27054379) have provided a quantitative model of how Glu impacts protein stability in relation to the amount and composition of surface area exposed or buried. While the authors cite these studies, they contain much more relevant information than presently exploited by the authors. In particular, the authors should study the predictions of the model by Record et al. in the case of FUS sequence and critically analyze what they imply about their findings.

5. What is the contribution of protein net charge to the observed effects? E.g. Figure 2. g-i the net charges of the constructs are 10, -1, 12, respectively. As Glu is both a pi system and an anion, are there any implications concerning the anion-pi interactions?

6. On p. 4, the authors claim that "In KGlu, the slow modes appear at protein concentrations of 2 μM (Supplementary Figure 2e), whereas in KCl they appear at 3 μM . These results provide independent confirmation of the estimates of c_{sat} and emphasize the small differences in driving forces for macrophase separation of FUS-SNAP in 100 mM KCl versus 100 mM KGlu".

How significant is this difference? Also, there does not appear to be a strong difference between panels in Supplementary Figures 2a and g. Moreover, since the sample is polydisperse, were there any regularization fits used to obtain size distributions? Also, the effect on the autocorrelation curves looks more prominent in panel c and not so in f. Since DLS has a detection limit, (e.g. 0.5 mg/ml or so), autocorrelation curves should also be presented for low protein concentrations (below 1 μM).

7. On p. 7, the authors claim that "the derived count rate for FUS-SNAP is up to 3-fold higher in 100 mM KGlu when compared to 100 mM KCl (Fig. 2g). Untagged FUS shows similar behaviors (Supplementary Figure 3g)." This is not immediately apparent, especially when it comes to quantitative differences. Also, the effect is not so prominent for another construct (FUS-EGFP). Finally, different constructs also have different net charges FUS WT 14, FUS-SNAP 10, FUS-EGFP 6. How does this impact the comparison?

8. The claimed difference between Arg and Lys residues should be more precisely phrased and better explained in light of the existing literature on the topic (p. 8 "Indeed, it is noteworthy that the free energies of solvation of Arg and Lys are fundamentally different from one another. From this hydration perspective, Arg has more of a hydrophobic character than Lys. As a result, the driving forces for phase separation, which are governed mainly by solubility, are affected by the substitution of Arg to Lys significantly.")

9. On p. 6, the authors claim that "Further, measurements of fluorescence correlation spectroscopy (FCS) on the single-molecule level show that the autocorrelation of donor-labelled FUS-SNAP is consistent with increased translational diffusion time τ_D and strong fluctuation in the weighted residuals at correlation times longer than τ_D in the KGlu buffer (Supplementary Figure 3f). This points to the presence of higher-order complexes even at 400 pM concentrations of FUS-SNAP." However, it is hard to see any difference. This should be shown quantitatively together with an estimation of how significant the effect is. Also, since here one again deals with polydisperse media, how appropriate is it to just use a two-component fit?

10. Some ring-containing amino acids (at least Tyr) should be added to the list of those studied by MD simulations, especially considering their importance in condensate formation. Moreover, some RDFs (SI Fig. 5.) from MD simulations do not seem to fully converge at the studied distances (e.g. Lys or Arg). This should be extended.

Minor comments

1. L64. Remove "of"
2. L279. Remove the second instance of "are affected"
3. L353. For simplicity, the order of amino acids should correspond to the order given in figure 6c.
4. In MD simulations, the authors interchangeably use "amino acid" and "peptide" to refer to capped amino acids. For clarity, it may be good to just use "capped amino acid" or similar, although the species is technically a peptide.
5. L689 explain what is meant by "scale factor"?
6. L691. express the salt concentration in units of mol of the anion.
7. Fig 6 a-c legend. It should be indicated in the legend that these are MD simulation results.
8. Figure 2 caption: "DLS data shows the derived count rate of FUS-SNAP (d), TAF15-SNAP (e), and EWSR1-SNAP (f) in different buffers". The labels should be corrected (g-i).

Responses to Reviewers

We thank the reviewers for their comments and assessments of our work. We have made extensive revisions that include new results, additional analyses, an expanded introduction, and discussion section. These revisions are fully responsive to the comments made by the reviewers. In the revised version of the manuscript, the text that has changed significantly has been marked in red. In the point-by-point responses, we have transcribed sections of the revised text to demonstrate responsiveness to the specific comments and concerns raised by the reviewers. The responses to reviewers are formatted as follows: The comments made by the reviewers are in italics, our responses are in normal text, and any transcribed material from the revised manuscript is in red.

Responses to Reviewer 1

Summary comments: *In this manuscript, the authors investigated the effects of glutamate versus chloride on both macrophase separation and clustering in sub-saturated solutions of FUS and specific FET family proteins. It proved that the macrophase separation and clustering of the proteins are two distinct and separable processes, by using two deferent salts (glutamate and chloride) as the background. This manuscript can be improved on following aspects.*

Response to summary comments: We thank the reviewer for this assessment. We have made extensive revisions and provided detailed responses to the comments made by both reviewers.

Comment 1: *The conclusion of this paper is similar to the previous work of the authors (reference 7) but different methods were used. Therefore, the authors need to state the scientific novelty and its difference with the previous work.*

Response to comment 1: As noted in the summary statement provided by the reviewer, the objective of the current work was two-fold: First, we assess whether the interactions that drive percolation or pre-percolation versus phase separation are separable. We show that this is indeed the case. Second, and most importantly, glutamate, unlike chloride, is an exemplar of the anions found within cells. Therefore, establishing that FET proteins show enhanced clustering in glutamate when compared to chloride is both novel and highly relevant for understanding the functions of these proteins in cells. The revised introduction includes the following justification / motivation for the current study. Please see the text, which starts on line 96 of the revised manuscript:

Extant data suggest that the sequence determinants of clustering in sub-saturated solutions and the solution-condition-specific values of c_{sat} can be both synergistic and distinct⁴². For example, several mutations with FET proteins were found to affect clustering and phase separation equivalently. However, separability of interactions was also demonstrated by the effects of solutes that dissolve condensates but do not influence clustering in sub-saturated solutions⁴². Likewise, while certain mutations impact clustering, they have a minimal impact on phase separation, especially if c_{sat} is already low. The recent study of Lan et al. reported findings for Negative Elongation Factor (NELF) that support the separation of interactions that determine sub-saturated solution clusters versus condensation in live cells⁴³. These observations suggest that, in some systems, clustering and phase separation are likely to be governed by partially separable energy scales. Here, we investigated whether changing the solution anion from chloride to glutamate would have separable effects on the driving forces for phase separation versus clustering in sub-saturated solutions.

Our choice of comparing the effects of chloride versus glutamate on clustering versus phase separation was motivated by two considerations: First, glutamate tends to drive protein-protein associations^{44,45}. This, we reasoned, should enhance clustering in sub-saturated solutions. Second, cellular milieus are complex mixtures of ions, metabolites, and osmolytes^{46,47}. The high concentrations of potassium (~150 mM) are balanced by anions that include the amino acid glutamate, glutathione, and organic phosphates^{47,48}. Importantly, the relevant anion inside cells is glutamate and other organic phosphates whereas the intracellular concentrations of chloride are very low in comparison (see **Supplementary Table S1** for information regarding glutamate)^{46,47}.

Finally, we provide the following rationale, starting on line 140 of the revised manuscript.

We reasoned, based on our findings regarding the different effects of solutes on clustering versus phase separation of FET proteins⁴², that studying clustering in glutamate versus chloride would allow us to assess whether interactions that drive clustering / percolation are separable from those that drive phase separation.

In the revised discussion section, we summarize our results and provide comparative assessments with the work of Kozlov et al., that is also discussed extensively. Please see the following text, which starts on line 513 of the revised manuscript:

In the mean-field formalism of Flory³² and Huggins³³, the solubility parameter χ is proportional to the algebraic difference between the effective protein-solvent interactions and the arithmetic mean of protein-protein and protein-solvent interactions⁵. For associative macromolecules, Tanaka introduced the concept of a renormalized χ to account for the effects of specific intermolecular associations, and the interplay with the solvent-specific interactions³⁶. The renormalized χ depends on macromolecular concentrations and it combines the contributions of specific sticker-stickers interactions, their mediation by solvent, and the effects of the interplay between solubility-determining macromolecule-solvent interactions and solvent-solvent interactions. In glutamate, our data show that associations of FET proteins are strengthened on the nanoscale. However, these do not translate to significant changes in c_{sat} . The implication is that, for FET proteins, the renormalized χ is similar in chloride versus glutamate. This suggests that the enhancement of protein-protein interactions on nanoscales is counterbalanced by length-scale-dependent changes to macromolecule-solvent interactions and / or weakened solvent-solvent interactions in glutamate versus chloride. This would explain why cluster formation in sub-saturated solutions is enhanced, but c_{sat} changes only minimally.

Our findings regarding the relative insensitivity of c_{sat} to glutamate versus chloride differ from those of Kozlov et al. for bacterial SSBs⁵⁴. This suggests that the interplay of solvent-mediated specific associations and solubility-determining interactions are not generic across different systems, but they are instead sequence- and architecture-specific. While FET proteins are flexible, linear associative polymers, the SSBs are protein-based exemplars of branched “hairy colloids”^{70,71}. Taken together with the results of Kozlov et al.,⁵⁴ our work highlights the need for comparing the effects of glutamate and other cellular metabolites on clustering and phase separation of multivalent proteins defined by different sequence grammars and architectures.

Our comparative assessments of chloride versus glutamate were motivated by the fact that the latter is an exemplar of the types of anions that are present in cells. Yet, it is often assumed that physiologically relevant salt conditions correspond to 100 - 150 mM KCl or NaCl^{72,73,74,75,76}. This assumption does not square with extant data for prokaryotic⁴⁷ or eukaryotic systems. For example, in the cytoplasm of glutamatergic neurons, the concentration of glutamate is in the 5 – 10 mM range⁷⁷. In synaptic vesicles, glutamate concentrations can be as high as 100 mM⁷⁸. Formulations for a “single-assay medium” intended to mimic the *in vivo* medium of *Saccharomyces cerevisiae* lead to the following prescription: 300 mM K⁺, 245 mM glutamate, 50 mM phosphate, 20 mM Na⁺, 2 mM free Mg²⁺, all at a pH of 6.8⁴⁸. Importantly, the mimicking medium does not include chloride. Therefore, RNA-binding proteins are unlikely to encounter chloride inside cells. Instead, glutamate or other metabolites including phosphates are likely to be the key anions, thus highlighting the biological relevance of findings reported in this work.

Comment 2: *Introduction. Please give explanation to the fact that ATP and 1,6-hexanediol dissolve condensates while do not affect the clustering.*

Response to comment 2: This topic was discussed extensively in Ref. 42, which is the original work of Kar et al., and we do not discuss it here. To avoid distractions, we have deleted mentions of ATP and 1,6-hexanediol, since these solutes are not the focus of the current work.

Comment 3: *Page 9. The authors used two fluorescent probes, Nile red and bis-ANS. As Nile red shows increased quantum yields in nonpolar environments, what is the property of bis-ANS in response to its environment?*

Response to comment 3: We have rewritten the relevant section and added supplementary figures to provide an explanation of this issue. The following text, which starts on line 425, alongside **Fig. 6**, address the issue raised by the reviewer.

Clusters create unique local environments: Next, we used environmentally-sensitive dyes, specifically Nile red and bis-ANS, to probe the physicochemical properties of clusters that form in different buffers. The quantum yields of both dyes show increased quantum yields in nonpolar environments^{46,73,74}. We measured the fluorescence lifetime distributions of Nile red using MFD in various concentrations of FUS-SNAP in 100 mM KGlu and 100 mM KCl. Nile Red exhibits a spectrum of lifetimes ranging from 0.6 ns to 4.66 ns. It is known the lifetimes of Nile Red increase with increased hydrophobicity⁷⁵. In 100 mM KCl, at 0.5 μM FUS-SNAP, the peak in the fluorescence lifetime distribution occurs at 2 ns. With increasing concentration of FUS-SNAP, the lifetime distribution in 100 mM KCl becomes bimodal, showing peaks at 2 ns and 4 ns. In the presence of 100 mM KGlu, the Nile Red lifetime distributions show one distinct peak with an average lifetime of 4 ns, which is reached at

FUS-SNAP concentrations as low as $0.5 \mu\text{M}$. The inference is that there is an increase in the number and size of clusters in KGlu compared to KCl (**Fig. 7a**).

Fig. 7: FUS-SNAP clusters create distinct local environments. (a) The fluorescence lifetimes of Nile Red with various concentrations of FUS-SNAP equilibrated for 30 minutes in KGlu and KCl buffer. (b)-(d) The different concentrations of FUS-SNAP solutions and buffers are mixed with $2 \mu\text{M}$ bis-ANS in 100 mM KGlu buffer (b), 100 mM KCl buffer (c), and 200 mM KCl buffer (d). For bis-ANS studies (b)-(d), the mixture solutions were excited using a 355 nm laser, and the emission spectra were measured from 425 nm to 650 nm.

To complement the analysis with Nile Red, we also used bis-ANS to probe the local environments within clusters that form in the presence of 100 mM KGlu versus 100 mM KCl (**Fig. 7b** and **7c**). In both cases, in the presence of $2 \mu\text{M}$ bis-ANS concentration, the fluorescence intensity increases with increasing protein concentration. The increase in intensity is higher in the presence of KGlu compared to KCl. As a control, when we increased the KCl concentration to 200 mM, it caused a decrease in the fluorescence intensity of bis-ANS with FUS-SNAP compared to 100 mM KCl buffer. This suggests that KCl inhibits the clustering of FUS-SNAP. To assess the hydrophobicity of the clusters, the fluorescence intensity of the same concentration of bis-ANS was measured in methanol and ethanol (**Supplementary Fig. S7b**). The intensity of bis-ANS in the presence of FUS-SNAP clusters in KGlu buffer is comparable to that of bis-ANS in methanol. These findings suggest that clustering in sub-saturated solutions is enhanced in KGlu when compared to KCl. Stronger molecular associations increase the extent of clustering, as probed via the sizes of clusters, and the larger clusters are internally more hydrophobic when compared to the surrounding solvent.

Responses to Reviewer 2

Summary comments: Kar and coworkers present a combined experimental/computational analysis of the impact of glutamate anions, major constituents of the cytosol, on the formation of biomolecular condensates of protein FUS and other members of the FET protein family. As the main theme of the study, they compare and contrast the impact of glutamate with that of chloride ions within a framework in which percolation is coupled with phase transition in condensate formation. The question of the mechanistic foundation of biomolecular condensate formation is an important one and the authors

provide several relevant and thought-provoking results in this regard. However, there is a number of key issues that need to be addressed adequately before the suitability of the manuscript for publication in *Nature Communications* can be assessed.

Response to summary comments: We thank the reviewer for providing a detailed assessment of our work. Below, we provide a detailed point-by-point response and an inventory of the revisions we have made in response to the comments made the reviewer.

Comment 1: *A difficulty with reading the article comes from the fact that the authors do not provide a clear description of the expected quantitative, experimentally observable differences between phase separation and percolation processes. What does one expect if the observed clusters (meso- or macro-) are formed by separation or percolation? Does one expect a difference in cluster size distribution for the same concentration point? Intuitively, phase separation vs. percolation should behave quite differently since in the first case it is an all-or-nothing transition (no clusters observed until threshold concentration) and in the latter case a continuum of clusters should appear at concentrations below the percolation threshold.*

Response to comment 1: We have revised the introduction to provide a clear distinction of phase separation sans percolation versus phase separation coupled to percolation. The following revisions and inclusions are provided, starting on line 39 of the revised manuscript. Please note that this includes a new figure to answer the query as directly as possible.

In vitro, in the presence of 150 mM KCl and at a pH of ~ 7.2 , FET proteins purified from insect cells will undergo phase separation above sequence-specific saturation concentrations (c_{sat})¹⁰. The sequence-dependencies of measured c_{sat} values were rationalized using the framework of linear associative polymers¹², which can be used to parse the sequences of FET proteins into stickers versus spacers^{10, 13, 14, 15, 16, 17}. Stickers engage in strong, specific interactions, and they form reversible physical crosslinks with one another. Spacers contribute to the cooperativity of sticker-sticker interactions^{6, 16, 18}. They also contribute to macromolecular solubility through the balance of spacer-spacer, spacer-sticker, and spacer-solvent interactions^{19, 20, 21}. Distinct hierarchies of interactions that enable the classification of residues or motifs as stickers versus spacers enables the mapping of different heteropolymeric systems onto linear associative polymers. These include the intrinsically disordered RGG domains of DDX4 and LAF-1^{22, 23, 24}, full-length FET proteins¹⁰, their RBDs and PLCDs^{19, 20, 21, 25, 26}, the condensate driving domains of chromatin remodeling complexes²⁷, the stress granule protein UBQLN2²⁸, and unfolded states of intrinsically foldable domains¹⁷.

Phase transitions of associative macromolecules combine phase separation and percolation^{5, 29}. The latter is also known as thermoreversible gelation^{12, 13, 30, 31}. The solubility limits of macromolecules, influenced by solvent-mediated intermolecular interactions, will lead to the separation of a macromolecular solution into coexisting dense and dilute phases^{32, 33}. Equalization of chemical potentials and osmotic pressure³⁴ will determine macromolecular concentrations in dense and dilute phases, and we designate these as c_{den} and c_{dil} , respectively. Note that c_{dil} equals c_{sat} . Phase separation is a segregative transition because the macromolecular solution separates into coexisting dense and dilute phases to minimize the overall free energy of mixing of macromolecular and solvent components^{5, 35}.

Multivalence, defined by the numbers of stickers of different types, will enable the networking of associative macromolecules through the formation of clusters that grow continuously with increasing numbers of molecules being incorporated into networks as concentrations increase^{5, 13, 16, 36}. These continuous transitions are defined by a percolation threshold (c_{perc}) above which a system-spanning network forms^{5, 31, 36, 37, 38}. As clusters grow, they will influence solubility. This is because, as emphasized by Ogston³⁹, overall solubility is governed by a combination of the sizes and physicochemical properties of clusters that form via intermolecular associations³⁶. For associative macromolecules that undergo phase separation it follows that $c_{\text{sat}} < c_{\text{perc}} < c_{\text{den}}$ ^{5, 6, 13, 14}. As a result, associative macromolecules that undergo phase separation will also undergo percolation, with the dense phase being a physically crosslinked network that spans the condensate^{5, 21, 40}. This gives condensates an underlying network-like sub-structure that generates sequence-specific viscoelastic moduli⁴¹. Gelation or percolation sans phase separation will occur if $c_{\text{perc}} < c_{\text{sat}}$ ⁶.

A direct upshot of the coupling of phase separation and percolation is the presence of pre-percolation clusters in sub-saturated solutions³⁶. Theory predicts that the average cluster size will grow continuously with concentration, where sizes are defined by the numbers of molecules within clusters (**Fig. 1**)^{5, 36}. In accord with these expectations, recent studies, which deployed a diverse suite of measurements spanning a wide range of concentrations, showed that FET family proteins form heterogeneous distributions of pre-percolation clusters in sub-saturated solutions⁴². With increasing protein concentration, the clusters in sub-saturated solutions grow

continuously in size and abundance. The distributions of cluster sizes in sub-saturated solutions are heavy-tailed^{5, 42} (**Fig. 1**). This affords a finite likelihood of forming mesoscale species, hundreds of nanometers in diameter. The presence of these species contributes to saturating the soluble phase, thus contributing to the determination of c_{sat} . Indeed, as shown previously⁴², systems with weak clustering are also characterized by larger c_{sat} values. Above c_{sat} , condensate formation of FET proteins is driven by the separation of large and small species via cluster-cluster coalescence and the networking of mesoscopic clusters that form in sub-saturated solutions⁴².

Fig. 1: Cluster size distributions in sub-saturated solutions for different processes. If phase separation does not involve associative interactions and is driven by a single energy scale, *viz.*, macromolecular solubility, then the cluster size distribution will be bounded, as shown by the dashed line. However, if specific interactions between stickers contribute to associations, then the cluster size distribution evolves continuously, showing a rightward shift as c_{sat} is approached – see solid lines. The ordinate quantifies $P(n)$, the probability density associated with a cluster comprising n molecules, which is the label along the abscissa.

Extant data suggest that the sequence determinants of clustering in sub-saturated solutions and the solution-condition-specific values of c_{sat} can be both synergistic and distinct⁴². For example, several mutations with FET proteins were found to affect clustering and phase separation equivalently. However, separability of interactions was also demonstrated by the effects of solutes that dissolve condensates but do not influence clustering in sub-saturated solutions⁴². Likewise, while certain mutations impact clustering, they have a minimal impact on phase separation, especially if c_{sat} is already low. The recent study of Lan et al. reported findings for Negative Elongation Factor (NELF) that support the separation of interactions that determine sub-saturated solution clusters versus condensation in live cells⁴³. These observations suggest that, in some systems, clustering and phase separation are likely to be governed by partially separable energy scales. Here, we investigated whether changing the solution anion from chloride to glutamate would have separable effects on the driving forces for phase separation versus clustering in sub-saturated solutions.

Comment 2: *The experimental methods used in the study rely mostly on monitoring particle diffusion (e.g. DLS, FCS, NTA) and, as such, they report on the apparent sizes, but not the masses of cluster i.e. the number of proteins in them. It is important that the effects are clearly demonstrated for the mass distribution of clusters. For example, can the derived count rates in DLS data be converted to SLS intensities and be related to mass (Figure 2 g-i, Figure S3 g-h)? This is important as the observed changes in the apparent hydrodynamic size can have many different explanations, e.g. protein can become swollen in one buffer and compact in another and the same could translate to their respective oligomers. In other words, two clusters can contain the same number of proteins, while being of a different hydrodynamic diameter. Conversely, they can have the same diameter, while containing a very different number of proteins.*

Response to comment 2: As explained in response to **comment 3** (please see below), we have added new, single molecule measurements to test whether there are conformational changes that occur and are different between the two buffers. We do not find any evidence for the proposed conformational changes. Furthermore, such conformational changes would have to be quite significant for them to manifest as differences in hydrodynamic sizes in scattering measurements. Even more importantly, we

draw the reviewer's attention to the fact that the single molecule measurements – both the original data and the new data – which provide assessments regarding the numbers and brightness per molecule, irrespective of conformational changes, while also investigating the prospect of conformational changes, provide a clear denouement in favor of clustering rather than changes to scattering based purely on conformational changes. Indeed, the sizes of scatterers we observe in the NTA measurements would not be realizable based purely on conformational transitions in the single molecule regime. As for the request to convert the DLS data to SLS intensities, this can only be done if we had access to multi-angle DLS data. Here, the DLS data were collected at 173°. Likewise, for mass measurements, we would need a combination of static right-angle light scattering and low-angle light scattering or multi-angle light scattering. Even these methods, which will always be heavily influenced by Rayleigh scattering, will not be effective for deconvolution of mass distributions because our samples, as unequivocally shown by the single-molecule and multiparameter fluorescence data, have significant heterogeneities.

Comment 2: *In light of this, the authors should directly assess the conformational properties of single protein molecules in different environments.*

Response to comment 2: We have added a new section and a brand new, multi-panel figure (**Fig. 4**) that directly addresses this point. The relevant inclusions may be found starting on line 282 and are reproduced here for the benefit of the reviewer. Importantly, we have used sensitive measurements of conformation at ultra-low proteins concentrations and quantified the differences in cluster size distributions across a range of protein concentrations in the two buffers. We do not find any evidence of conformational differences between the two buffers.

FCS and NanoDSF also show enhanced stabilization of FUS clusters in glutamate: IUPRED analysis⁶⁰ predicts that isolated FUS mainly consists of disordered regions (**Fig. 4a**). However, given extant sequence-ensemble characterizations of disordered proteins, it stands to reason that there will be conformational fluctuations that are differently impacted by glutamate versus chloride. To investigate the influence of both buffers on the stabilization of FUS in monomers and in clusters, we studied FUS-SNAP-AF488 in a complementary approach by FCS at the single-molecule level and by nanoscale differential scanning fluorimetry (nanoDSF) at concentrations close to saturation where the signals will be dominated by larger, non-monomeric species.

We studied single-molecule events in equilibrated solutions with FUS-SNAP-AF488 in KGlu and KCl, respectively (**Fig. 4b**). Comparing both signal traces, it is evident, that the bursts in KGlu are brighter and more frequent. We computed the autocorrelation functions of FUS-SNAP-AF488 for two intensity-based selections: monomers and clusters. For the preferential selection of monomers, all bright bursts above its intensity threshold are excluded. In **Fig 4c**, we show correlation curves for FUS monomers together with the free dye measurement of rhodamine 110 as reference. The data for FUS were fit using a model (see Methods, eq. 6a) with two components for translational diffusion: one global time for dye impurities and one salt-dependent time for monomeric FUS. In the panels on the right, we show two blow-ups of the correlation curves centered at the respective diffusion times of FUS monomers, $t_{d,monomer}$, (dark yellow) and clusters, $t_{d,cluster}$ (pink). For FUS monomers, the correlation curves in KCl and KGlu overlap, and the fitted diffusion times are identical within error. Using the Stokes-Einstein equation, we converted these times to an average hydrodynamic radius of 2.3 nm. The distinct buffers do not change the overall size of monomeric FUS.

In contrast to the monomer sub-population, the long diffusion time $t_{d,cluster}$ (**Fig. 4d**) of FUS clusters in KGlu differs from the value in KCl by ~ 1 ms (**Supplementary Table S4**). Furthermore, large deviations in the weighted residuals indicate the insufficiency of a two-component fit for FUS clusters in glutamate. Thus, we applied the Maximum Entropy Method (MEM) as a model free approach^{61, 62} to quantify the diffusion time distributions for clusters (**Fig. 4e**). Two peaks were obtained for both buffers. To verify the goodness of the fit and demonstrate appropriate weighting, we display the dependence of the reduced χ^2_{red} on the entropy (L-curve, **Fig. 4f**), where the chosen values of the corner point are marked with a dot (see Methods and **Supplementary Fig. S6**). Due to the larger fraction of clusters, a higher minimum χ^2_{red} is yielded for KGlu. The first peak at $t_{d,monomer}$ ~ 0.2 ms resembles the monomer species and they overlap for KCl and KGlu. The second peak, which is in the millisecond time range, corresponds to clusters. In KGlu, the peak has significantly longer times and higher amplitudes than in KCl. From this we conclude that FUS clusters are more abundant and larger in size in KGlu, even though there are no detectable conformational differences at the level of FUS monomers. Instead, glutamate enhances macromolecular associations when compared to chloride, and this is in line with the previous reports^{50, 55}.

Glutamate is known to enhance protein stability⁴⁵. Although FUS is intrinsically disordered, its overall dimensions and the heterogeneity of intramolecular interactions, quantified via accessibility of different functional groups, will be temperature dependent. Accordingly, we investigated how buffers influence the temperature dependence of tryptophan fluorescence of FUS-SNAP. For this, we performed nanoDSF measurements, which helps us analyze the consequences of thermal fluctuations in low-volume capillaries. Increasing the temperature will drive increased exposure of tryptophan residues. NanoDSF monitors the concurrent changes in tryptophan fluorescence at 330 and 350 nm⁶³. To increase the measurement sensitivity, we used FUS-SNAP instead of FUS. This helps minimize the amount of residual KCl caused by the storage buffer, and it enables improved signal strength.

Fig. 4g shows changes of the 350 nm / 330 nm ratio as a function of increasing temperature in two different concentrations and buffers. The first derivative plot (**Fig. 4h**) shows that the apparent unfolding temperature of FUS-SNAP in the K₂Glu and KCl buffer is 57°C and 53°C, respectively. We also tested the SNAP-tag alone as a control. The apparent unfolding temperature of SNAP in the K₂Glu and KCl buffer is 65°C and 69°C, respectively (**Supplementary Fig. S6**). Surprisingly, in the K₂Glu buffer, SNAP has lower apparent unfolding temperature than in the KCl buffer. Therefore, the enhanced thermal stability of FUS-SNAP in glutamate can be attributed to FUS and not to SNAP. Taken together, the FCS and nanoDSF measurements demonstrate that glutamate enhances intramolecular and intermolecular interactions among FUS molecules when compared to KCl.

Fig. 4: Compared to KCl, K₂Glu stabilizes FUS-SNAP clusters even at ultra-low concentrations. (a) IUPRED predictions of disorder of FUS-SNAP. (b) Single-molecule detection (SMD) fluorescence intensity traces of 200 pM FUS-SNAP-AF488 in both buffers with indicated threshold for cluster cut (yellow-green) and monomer reference cut (pink). These traces display the pronounced clustering behavior in K₂Glu compared to KCl. (c-d) Autocorrelation curves for FUS-SNAP-AF488 with blow-ups for the monomer time window (yellow-green) and oligomer time window (pink) show the resulting translational diffusion times (dashed lines) including one global and buffer-dependent time for a 3D-Gaussian diffusion model and the weighted residuals for the cluster cut and the monomer cut as reference. The fit to eq. 6a (see Methods) in panel c yields identical diffusion times within error in KCl ($t_{d,mo}^{(KCl)} = 0.189 \pm 0.017$ ms, orange) and K₂Glu ($t_{d,mo}^{(KCl)} = 0.201 \pm 0.017$ ms, blue (for all fit results see

Supplementary Table S4). Additionally, the free dye measurement of Rhodamine (Rh110) is given as reference (grey) in panel c. The correlation curve for the cluster cut displays in KGlu (dark blue) a clear shift to longer diffusion times in the oligomer time window when compared to KCl (red). The monomer component (dashed black) is adequately fitted with one diffusion time for both buffers (see **Supplementary Table S3**). (e) The maximum entropy method (MEM) gives diffusion time distributions (101 components) as a function of probability with one peak between 0.09 and 0.4 ms (monomer time window, green) and a second peak between 0.8 and 4 ms (oligomer time window, pink). (f) Corresponding L-curves according to Vinogradov et al.,⁶² are presented as quality validation for the obtained diffusion time distributions where the corner point is indicated by a circle. (panels g-h). NanoDSF data show the ratio of 350 nm/330 nm plotted against temperature for FUS SNAP in KCl and KGlu buffers. (h) The first derivative of data shown in (g) against temperature shows the apparent unfolding temperature of FUS-SNAP at 57°C and 53°C in KGlu and KCl buffers, respectively.

Comment 3: *Importantly, in the absence of a precise estimation of the size of the clusters in terms of mass or number of proteins, any thermodynamic arguments or estimates related to the free energy of transfer of FUS molecules from the dispersed phase into mesoscale clusters in glutamate versus chloride cannot be made. For example, on L172-173, the authors estimate that this difference is 0.5-0.75 kcal/mol, but this must come from a tacit assumption that the abundance of mesoscale clusters is a direct measure of the amount of protein molecules contained in them, which need not be the case. As already mentioned, the authors should measure the sizes of clusters in terms of the mass of proteins they contain or, at least, clearly state and critically discuss the assumptions behind connecting cluster sizes with protein mass. In general, the authors should from the start precisely define what they mean by “size of clusters” throughout. This is not just a semantic point, but rather concerns the very essence of the arguments made.*

Response to comment 3: By cluster size, we refer to the number of molecules per cluster. This point has been spelled out very clearly. This is perfectly valid, since there is no evidence that there are drastic conformational changes across the two buffers, and the fluorescence data are unambiguous in terms of the origins of clustering between intermolecular associations as opposed to intramolecular conformational changes. However, the cluster distributions are heterogeneous, and we do not have precise values for $P(n)$ as a function of n . We have used the maximum entropy method to estimate this distribution, but it is an estimate. Therefore, we have deleted any mentions of transfer free energies.

Comment 4: *Record and coworkers (PMID: 27806267, 27054379) have provided a quantitative model of how Glu impacts protein stability in relation to the amount and composition of surface area exposed or buried. While the authors cite these studies, they contain much more relevant information than presently exploited by the authors. In particular, the authors should study the predictions of the model by Record et al. in the case of FUS sequence and critically analyze what they imply about their findings.*

Response to comment 4: The formalism of Kirkwood and Buff is the most rigorous description of solution thermodynamics in ternary mixtures and of preferential interactions, which are quantified by the coefficients that we calculate from simulations. In accord with the parsing of Record and coworkers, we conclude that the associations of FET proteins appear to be driven by the preferential exclusion of glutamate when compared to chloride. However, our site-specific radial distribution functions, detailed in **Supplementary Figs. S9 – S12**, show that there are discrepancies between inferences drawn by Record and co-workers and our computations for sp^2 and cationic nitrogen atoms. We have added a detailed discussion of this issue. Please see the revised discussion section, especially the section that starts on line 550, which we reproduce below.

We used molecular simulations to quantify preferential interaction coefficients for KCl and KGlu around amino acids with different sidechain chemistries. In line with the proposals of Record and coworkers^{44, 45, 53, 54}, our simulations show that glutamate is preferentially excluded from backbone and sidechain amides, as well as other functional groups. However, there are key differences in the atomic-level details that emerge from our simulations versus interpretations proposed by Record and colleagues^{44, 45, 53, 54}.

Cheng et al., used vapor pressure osmometry (VPO) to measure the solubilities of model compounds in aqueous solvents with different types of solutes⁴⁴. In their notation, water, the primary component is labeled 1, the model compound of interest is component 2, and the solute of interest, such as KGlu or KCl, is component 3. The change in solubility, as gleaned from VPO measurements, leads to inferences regarding the sign and magnitude of the chemical potential μ_{23} for the preferential interaction of the model compound with the solute. A positive sign indicates preferential exclusion, whereas a negative sign implies preferential interactions. The

measured μ_{23} values for fifteen different model compounds were decomposed using a global regression analysis based on a linear superposition model⁴⁴. This model is written as: $\mu_{23} = \sum \alpha_i A_i$. Here, the summation is over atoms of functional groups in the model compounds and A_i is the solvent accessible surface area of atom i within the model compounds. The values of A_i are computed using a specific probe radius using structures for each model compound. Cheng et al. derived values of α_i from a global regression analysis of μ_{23} values for fifteen model compounds. Based on the inferred values of α_i , interactions of glutamate are proposed to be favorable for sp^2 nitrogen atoms and the nitrogen atoms of cationic residues. The converse was found to be true for chloride. Radial distribution functions from our simulations suggest the opposite trends when compared to the inferences of Cheng et al. (**Supplementary Figs. S9-S12**). For sp^2 oxygen and backbone carbon atoms, Cheng et al.,⁴⁴ inferred weaker interactions with chloride when compared to glutamate. Our simulation results show similar trends (**Supplementary Figs. S9-S12**).

What might be the source of discrepancies between inferences from analysis of VPO measurements versus results from simulations for sp^2 nitrogen atoms and the nitrogen atoms of cationic residues? Unlike the analysis of Cheng et al.,⁴⁴ the inferences from simulations were derived via a detailed accounting of the interplay of amino-acid, water, and solute interactions. The experimental data report one number for each model compound, and these are then dissected using an accessible surface area-based model combined with global regression analysis. There has been considerable debate regarding the use of solvent accessible areas for parsing measurements and drawing inferences regarding solvation thermodynamics^{79, 80}. The gist is that the use of accessible surface area as a measure of solvation is problematic for small molecules and atomic-level dissections. Accessible surface area only becomes a useful proxy only at larger length scales^{80, 81}. This is because the concept of an interfacial tension does not apply on the atomic and molecular length scales. Instead, theory and simulation suggest that the hydration thermodynamics and forces require the inclusion of a volume term and dispersion interactions on atomic and molecular scales⁸². These are fully present in our simulations. Additionally, solvent accessible surface area is insensitive to changes in conformation and the local concentrations of functional groups for linear, flexible systems⁸³. Hence, while the use of solvent accessible surface area is widely prevalent in the protein folding literature, and has been justified by elegant connections to the rigorous Kirkwood-Buff integrals⁸⁴, their use for dissecting atomic-level interactions of disordered proteins remains questionable. The preferential interaction coefficients we compute are directly gleaned from pair distribution functions, in accord with the Kirkwood-Buff formalism⁸⁵. At this juncture, we lean on consistencies of interpretations from the work of Cheng et al.,⁴⁴ and the simulations. Both sets of studies suggest that the central differences between chloride and glutamate derive from the latter being preferentially excluded from protein sites.

Comment 5: *What is the contribution of protein net charge to the observed effects? E.g. Figure 2. g-i the net charges of the constructs are 10, -1, 12, respectively. As Glu is both a pi system and an anion, are there any implications concerning the anion-pi interactions?*

Response to comment 5: We draw the reviewer's attention to data for the variant of FUS where we replace 10 Asp residues and 4 Glu residues to Gly (10D/4E-G). If there are anion- π interactions, then they have a destabilizing effect, because as noted in the text, clusters with this variant are readily detectable in the nanomolar concentration regime, whereas mutations of the aromatic residues within the RBD of FUS require concentrations of 10 μ M for clusters to be observed.

Comment 6: *On p. 4, the authors claim that "In KGlu, the slow modes appear at protein concentrations of 2 μ M (Supplementary Figure 2e), whereas in KCl they appear at 3 μ M. These results provide independent confirmation of the estimates of c_{sat} and emphasize the small differences in driving forces for macrophase separation of FUS-SNAP in 100 mM KCl versus 100 mM KGlu". How significant is this difference? Also, there does not appear to be a strong difference between panels in Supplementary Figures 2a and g. Moreover, since the sample is polydisperse, were there any regularization fits used to obtain size distributions? Also, the effect on the autocorrelation curves looks more prominent in panel c and not so in f. Since DLS has a detection limit, (e.g. 0.5 mg/ml or so), autocorrelation curves should also be presented for low protein concentrations (below 1 μ M).*

Response to comment 6: We provide the numbers for c_{sat} measured using the Bradford assay and inferred using the DLS data. They are consistent with one another. With the Bradford assay, we estimate c_{sat} values of 2 μ M in KGlu and 3 μ M in KCl. The autocorrelation curves show that slow modes appear at 2 μ M in KGlu and at 3 μ M in KCl. Two independent methods yield consistent inferences. The revised supplementary materials include data for lower concentrations as requested by the reviewer. Please see **Supplementary Fig. S2**, which we reproduce below.

Supplementary Fig. S2: KGLu buffer minimally affects the driving forces for phase separation of FUS-SNAP, although the evolution of condensates is discernibly different. (a)-(c) The correlation function from DLS of solutions containing different concentrations of FUS-SNAP, 1 μM (a), 2 μM (b), and 3 μM (c) in 20 mM Tris.HCl, pH 7.4, with a final concentration of 100 mM KCl. Panels (d), (e), and (f) show the correlation functions of solutions containing 1 μM , 2 μM , and 3 μM concentrations of FUS-SNAP, respectively, in 20 mM TRIS.Glu, pH 7.4, with 100 mM KGLu. Panels (g), (h), and (i) show the correlation functions of solutions containing 1 μM , 2 μM , and 3 μM concentrations of FUS-SNAP, respectively, in 20 mM TRIS.HCl, pH 7.4, with 100 mM KCl. The total concentration of KCl is marked on the panels. The correlation coefficient values indicate the abundance of clusters in the solutions. The time axis correlates with the size of the species; larger sizes require more time to decay, and vice versa.

Comment 7: On p. 7, the authors claim that "the derived count rate for FUS-SNAP is up to 3-fold higher in 100 mM KGLu when compared to 100 mM KCl (Fig. 2g). Untagged FUS shows similar behaviors (Supplementary Figure 3g)." This is not immediately apparent, especially when it comes to quantitative differences. Also, the effect is not so prominent for another construct (FUS-EGFP). Finally, different constructs also have different net charges FUS WT 14, FUS-SNAP 10, FUS-EGFP 6. How does this impact the comparison?

Response to comment 7: We summarized the data as we see them. As the reviewer likely appreciates, DLS is sensitive only to mesoscale clusters. This is why we have gone to great lengths to quantify cluster size distributions, in unprecedented ways, using MFD, MCS, FCS, and FRET. We obtain a consistent picture across all modalities. Given the heterogeneous nature of the cluster distributions, the analysis is necessarily semi-quantitative in that we quantify what we observe, but we do not have n -resolved distributions for $P(n)$, which appears to be what the reviewer is seeking. Please note that no amount of parsing of DLS data will give us what the reviewer seeks because such data are blind to the

gamut of species that contribute to the cluster size distributions – defined as the number of molecules per cluster.

Comment 8: *The claimed difference between Arg and Lys residues should be more precisely phrased and better explained in light of the existing literature on the topic (p. 8 "Indeed, it is noteworthy that the free energies of solvation of Arg and Lys are fundamentally different from one another. From this hydration perspective, Arg has more of a hydrophobic character than Lys. As a result, the driving forces for phase separation, which are governed mainly by solubility, are affected by the substitution of Arg to Lys significantly.")*

Response to comment 8: We urge the reviewer to read the sentences in question in the context of the paragraph in which they are incorporated. Here is the relevant paragraph, which starts on line 381. We have asked colleagues to read this for us, and we do not understand what is imprecise about our verbiage. Arg is more hydrophobic than Lys, and this point has been made most unambiguously by the analysis published by Fossat et al., which we cite in our manuscript.

Substituting 24 Arg residues in the RBD to Lys increases c_{sat} by an order of magnitude compared to wild-type FUS¹⁵. Strikingly, while these substitutions weaken clustering in the presence of 100 mM KCl (**Fig. 5c**), the extent of clustering we observe in the presence of 100 mM KGlu is similar to that of wild-type FUS (compare **Fig. 5c** to **Supplementary Fig. S5g**). As shown in recent single-molecule studies, there is a uniform weakening of cation- π interactions in chloride salts⁷¹. In contrast, in glutamate, the differences between wild-type FUS and the 24R-K variant seem to be length-scale dependent. Specifically, while clustering, which is mainly impacted by cation- π interactions, is preserved upon substituting Arg to Lys, phase separation, which should be governed mainly by solubility, is weakened by Arg to Lys substitutions. This can be rationalized if the strengths of cation- π interactions are minimally affected by glutamate compared to chloride. However, the increase in c_{sat} points to differences in solubility driven by substitutions of Arg to Lys. Indeed, it is noteworthy that the free energies of solvation of Arg and Lys are fundamentally different from one another⁷², and Arg has more of a hydrophobic character than Lys⁷². As a result, the driving forces for phase separation, which are governed by solubility limits, are affected by substitutions of Arg to Lys, whereas clustering in sub-saturated solutions is not affected, especially in glutamate.

Comment 9: *On p. 6, the authors claim that "Further, measurements of fluorescence correlation spectroscopy (FCS) on the single-molecule level show that the autocorrelation of donor-labelled FUS-SNAP is consistent with increased translational diffusion time td_2 and strong fluctuation in the weighted residuals at correlation times longer than td_2 in the KGlu buffer (Supplementary Figure 3f). This points to the presence of higher-order complexes even at 400 pM concentrations of FUS-SNAP." However, it is hard to see any difference. This should be shown quantitatively together with an estimation of how significant the effect is. Also, since here one again deals with polydisperse media, how appropriate is it to just use a two-component fit?*

Response comment 9: We direct the reviewer's attention to the new **Fig. 4** and the new **Supplementary Fig. S6**. Please also see the detailed response to **comment 2**.

Comment 10: *Some ring-containing amino acids (at least Tyr) should be added to the list of those studied by MD simulations, especially considering their importance in condensate formation. Moreover, some RDFs (SI Fig. 5.) from MD simulations do not seem to fully converge at the studied distances (e.g. Lys or Arg). This should be extended.*

Response to comment 10: The requested additions and extensions have been made. Please see the revised **Fig. 8** and the new **Supplementary Figs. S9-S12**.

REVIEWER COMMENTS

Reviewer #1 (Remarks to the Author):

All the questions of the reviewers have been addressed. The current version is acceptable.

Reviewer #2 (Remarks to the Author):

The authors have addressed some of the comments raised in the original review, but the main issue unfortunately remains i.e. the absence of strong evidence concerning cluster size distributions in terms of the number of molecules i.e. mass. In the new Figure 1, the authors themselves provide a sketch of the key differentiating feature between phase separation without percolation and phase separation involving associative interactions. Unfortunately, their data does not provide hard evidence concerning this very feature i.e. $P(n)$. Some new evidence in this direction is given (e.g. new Figure 4), but it is in my opinion not conclusive. I fully agree with the authors that DLS and other techniques used in the study cannot provide n -resolved distributions of $P(n)$, but in the absence of such information, most of the key statements remain inconclusive. As the authors themselves say in response to Comment 7, their analysis is semi-quantitative, but to correctly test the predictions of their model, one needs fully quantitative evidence. For this reason, I find publishing of the work in the present form to be premature.

Responses to Reviewers

Responses to comments of Reviewer 1

Comment: All the questions of the reviewers have been addressed. The current version is acceptable.

Response to comment: We thank the reviewer for this assessment.

Responses to comments of Reviewer 2

Comment: The authors have addressed some of the comments raised in the original review, but the main issue unfortunately remains i.e. the absence of strong evidence concerning cluster size distributions in terms of the number of molecules i.e. mass. In the new Figure 1, the authors themselves provide a sketch of the key differentiating feature between phase separation without percolation and phase separation involving associative interactions. Unfortunately, their data does not provide hard evidence concerning this very feature i.e. $P(n)$. Some new evidence in this direction is given (e.g. new Figure 4), but it is in my opinion not conclusive. I fully agree with the authors that DLS and other techniques used in the study cannot provide n -resolved distributions of $P(n)$, but in the absence of such information, most of the key statements remain inconclusive. As the authors themselves say in response to Comment 7, their analysis is semi-quantitative, but to correctly test the predictions of their model, one needs fully quantitative evidence. For this reason, I find publishing of the work in the present form to be premature.

Response to comment: We thank the reviewer for pushing us to think harder about our data. We realized, after evaluating the reviewer's comment, that we have the data to assess how the cluster size distribution evolves. We certainly know this for the hydrodynamic diameter (d_H). We know the abundance of the mesoscale clusters from NTA data, and we can use the intensity data and information regarding time correlation functions to extract the distributions of d_H values. Furthermore, our single molecule measurements show that FUS does not undergo detectable changes in conformation. Therefore, we were able to use the distributions of d_H values from DLS data and convert these into estimates of $P(n)$ or $f(n)$ where f is the raw frequency. We have done this in the revised version. Please see the new **Fig. 4** and the revised text that starts on line 245. For completeness the figure and the text are reproduced below. We hope this satisfies the reviewer, because it fully addresses the persistent concerns expressed by the reviewer.

Size distributions of low abundance mesoscale clusters from analysis of DLS data: The mesoscale clusters represent 0.1% – 1% of all species that are present in sub-saturated solutions. The abundance of mesoscale clusters is higher in glutamate than in chloride, especially well below c_{sat} (**Fig. 3a-3c**). We used the number density of scatterers, extracted from the DLS data, and quantified the distribution of hydrodynamic diameters (d_H) of mesoscale clusters. We used this analysis to ask and answer the following questions: On the manifold of mesoscale species that are the least abundant, albeit largest species that form in sub-saturated solutions, what is the frequency of observing specific d_H values? If we use this distribution to estimate the molecularity distributions, which refer to the frequency or probability of observing mesoscale clusters with n molecules, what types of distributions do we obtain? Specifically, is there evidence for continuous evolution of the heavy tail in the cluster size distribution as depicted in **Fig. 1**, and is this evolution different in KCl versus KGlu? To answer these questions, we leveraged the fact that information regarding the time correlation functions combined with information regarding raw intensities can be used to extract distributions of scattering intensities, which scale as the sixth power of the

hydrodynamic diameter d_H . Using the Stokes-Einstein formula, these intensity distributions can be used to estimate the number densities of d_H values. Following the approach of Cohan et al.,⁶⁸ the intensity distributions were converted to distributions of d_H values using practical implementations of Mie scattering theory⁶⁶.

Fig. 4: Size and molecularity distributions of low abundance, mesoscale clusters. We extracted the distributions of d_H values for FUS-SNAP at different protein concentrations, all of which were in the sub-saturated regime. The top row panels (a) – (d) show the distribution of d_H values extracted in K₂Glu (dotted curves) and KCl (solid curves) for solutions with protein concentrations of 0.125 μ M (a), 0.25 μ M (b), 0.5 μ M (c), and 1 μ M (d). These distributions are shown as raw histograms, and hence the ordinate shows frequencies i.e., the number of occurrences of a d_H value between d_H and $d_H + \Delta$, where $\Delta = 0.1$ nm. In each panel, the abundance of species being analyzed is shown in the legend, and these values were extracted from NTA data shown in **Fig. 3b**. Rows 2 and 3 show the molecularity distributions, where molecularity refers to the number of molecules n within a cluster of size d_H . These distributions were computed by assuming that the molecules within clusters are spheres. The packing fraction can be set to be $p = 0.64$, for random close packing of spheres, panels (e)-(g) or $p = 0.33$, panels (h) – (k), assuming a packing density concordant with reports of the volume fractions of protein versus solvent in single protein condensates^{25, 26, 27}. In each panel, the solid curve corresponds to KCl, and the dotted curve corresponds to K₂Glu. The concentration of [FUS-SNAP] for each column of plots is shown at the top.

We extracted distributions of d_H values for FUS-SNAP at different protein concentrations. All measurements were performed in sub-saturated solutions. We compared the distributions in 100 mM KCl versus 100 mM K₂Glu (**Fig. 4a-4d**) at different protein concentrations. For three of the four protein concentrations (0.125 μ M, 0.25 μ M, and 0.5 μ M) the size distributions in K₂Glu are shifted to higher values when compared to KCl. The distributions in K₂Glu and KCl show the heavy tail nature that we anticipate from theory (**Fig. 1**). This point becomes clear when one accounts for the abundance of the mesoscale clusters, which we measured using NTA.

Interestingly, the continuous growth of mesoscale species with increased protein concentrations plateaus in KGlu as c_{sat} is approached. As a result, in a 1 μM protein solution, the distribution of d_{H} values in KCl catch up with the distributions in KGlu. This observation helps explain the similarities of c_{sat} values that we estimated in KCl and KGlu. It is also in line with the theories^{46, 50}, where the entropy of mixing becomes considerably less favorable as molecular or cluster sizes decrease – a phenomenon referred to as an entropic sink by Bracha et al.,⁶⁹.

Next, we converted the distribution of d_{H} values to estimate the molecularity distributions, i.e., the distribution of number of molecules within a mesoscale cluster. To extract these distributions, we use the fact that the d_{H} of monomeric FUS-SNAP is 4.6 nm (see below for a direct measurement). Assuming a spherical approximation for the monomers, the number of molecules n within a cluster of hydrodynamic diameter d_{H} can be estimated as: $n = p(v_{\text{c}}/v_{\text{m}})$. Here, v_{c} and v_{m} are the volumes of the cluster and monomer, respectively and p is a dimensionless packing fraction. The upper limit on p is 0.74 for crystalline packing of monomers within a cluster. If we assume random close packing of spheres, then $p = 0.64$. Conversely, if we assume that molecules are packed within clusters as they would be in dense phases, where the volume fraction of solvent is between 0.6 and 0.7, then we can set $p = 0.33$ ^{25, 26, 27, 32, 70}.

We analyzed the molecularity distributions for mesoscale clusters formed by FUS-SNAP by assuming two different values for p viz., 0.64 (**Fig. 4d-4g**) and 0.33 (**Fig. 4h – 4k**). Both assumptions paint a similar picture. The low abundance mesoscale clusters, which should be in the tails of the cluster size distribution, show a rightward shift toward larger numbers with increasing protein concentration. The cluster sizes in KGlu are larger than in KCl for three of the four concentrations. As c_{sat} is approached, the cluster sizes stop growing in KGlu, and the cluster size distribution in KCl becomes akin to what is observed in KGlu. When comparing these cluster size distributions to what we anticipate from theory (**Fig. 1**), it is important to remember that we are analyzing cluster sizes on the manifold of mesoscale species, whose abundance is low, in the range of 0.1% – 1%.

Summary comment: We understand that the introduction of new ideas is difficult to accept. With all due respect, we submit that the case we have presented for pre-percolation clusters, their adherence to the tenets of the physics of associative polymers, and the range of experiments and analyses we have brought to bear far exceeds anything that has been produced in the phase separation or condensate literature. We hope that the reviewer will see reason and grant that we have provided more than definitive evidence for our case.

REVIEWERS' COMMENTS:

Reviewer #2 (Remarks to the Author):

I appreciate that the authors have taken seriously the comments regarding the importance of experimentally determining the $P(n)$ distributions. This is not a minor point, but rather an essential element when discussing phase separation and percolation – for this reason, I agree with the authors' decision to include $P(n)$ distributions in Figure 4. The authors currently estimate $P(n)$ distributions by using the hydrodynamic radii of clusters, as obtained by DLS, together with the sizes of individual molecules and assuming different sphere packing models. One potential deficiency with this approach is that it assumes a scale-independent packing density, which may or may not be true, as is known for different disordered polydisperse media and colloid systems. The authors should provide a discussion of the potential shortcomings of their approach to estimate $P(n)$ in the discussion section. In addition, the authors should consider using a different terminology when referring to $P(n)$ other than "molecularity". Molecularity typically refers to the number of molecules that participate in a single-step chemical reaction e.g. "this reaction is bimolecular" and is, to the best of my knowledge, not standardly used when talking about cluster sizes in soft-matter literature. One option might be to refer to these distributions as "cluster size distributions". Related to this, the authors should use $P(n)$ notation in Figure 4 (as in Figure 1), or explain why they chose $f(n)$ there. With these minor changes, I find the manuscript to be acceptable for publication in Nature Communications.

Responses to reviewers

Comments of Reviewer 2

Comment: *I appreciate that the authors have taken seriously the comments regarding the importance of experimentally determining the $P(n)$ distributions. This is not a minor point, but rather an essential element when discussing phase separation and percolation – for this reason, I agree with the authors' decision to include $P(n)$ distributions in Figure 4. The authors currently estimate $P(n)$ distributions by using the hydrodynamic radii of clusters, as obtained by DLS, together with the sizes of individual molecules and assuming different sphere packing models. One potential deficiency with this approach is that it assumes a scale-independent packing density, which may or may not be true, as is known for different disordered polydisperse media and colloid systems. The authors should provide a discussion of the potential shortcomings of their approach to estimate $P(n)$ in the discussion section. In addition, the authors should consider using a different terminology when referring to $P(n)$ other than “molecularity”. Molecularity typically refers to the number of molecules that participate in a single-step chemical reaction e.g. “this reaction is bimolecular” and is, to the best of my knowledge, not standardly used when talking about cluster sizes in soft-matter literature. One option might be to refer to these distributions as “cluster size distributions”. Related to this, the authors should use $P(n)$ notation in Figure 4 (as in Figure 1), or explain why they chose $f(n)$ there. With these minor changes, I find the manuscript to be acceptable for publication in Nature Communications.*

Response to comment: We have detailed the methods used to estimate the cluster size distribution. We have stated our assumptions clearly, and we clarified that the assumption of a length-scale independent packing fraction may or may not be valid. We have also replaced the term molecularity with cluster size. Finally, we have explained why we use $f(n)$ rather than $P(n)$. As noted in the text, DLS does not probe the entirety of the size distribution. Therefore, a normalization without knowledge of the abundance of the small vs. large species, and the entirety of the species spectrum would be erroneous. Therefore, $f(n)$ is the correct way to depict the histogram of cluster sizes, which refers to the number of molecules within the cluster.

Please see the following text, which starts on line 272 of the revised manuscript.

We analyzed the distributions of cluster sizes for mesoscale clusters formed by FUS-SNAP by assuming two different values for p namely, 0.64 (**Fig. 4d-4g**) and 0.33 (**Fig. 4h – 4k**). Note that our choice ignores the possibility that the packing fraction might depend on the cluster size. Both choices for p paint a similar picture. The low abundance mesoscale clusters, which should be in the tails of the cluster size distribution, show a rightward shift toward larger numbers with increasing protein concentration. The cluster sizes in KGlu are larger than in KCl for three of the four concentrations. As c_{sat} is approached, the cluster sizes stop growing in KGlu, and the cluster size distribution in KCl becomes akin to what is observed in KGlu. When comparing these cluster size distributions to what we anticipate from theory (**Fig. 1**), it is important to remember that we are analyzing cluster sizes on the manifold of mesoscale species, whose abundance is low, in the range of 0.1% – 1%. Therefore, the DLS data help us probe the tails of the cluster size distributions. To go beyond the tails and obtain information regarding the totality of size distributions, we used multiparameter fluorescence measurements.